

# Capturing the interplay of dynamics and networks through parameterizations of Laplacian operators

Xiaoran Yan[1], Shang-hua Teng[2], Kristina Lerman[3] and Rumi Ghosh[4]

[1] School of Informatics and Computing, Indiana University, Bloomington, IN, United States
[2] Computer Science Department, University of Southern California, Marina Del Rey, CA, United States
[3] Information Sciences Institute, University of Southern California, Marina Del Rey, CA, United States
[4] Robert Bosch, LLC, Palo Alto, CA, United States

## ABSTRACT

We study the interplay between a dynamical process and the structure of the network on which it unfolds using the parameterized Laplacian framework. This framework allows for defining and characterizing an ensemble of dynamical processes on a network beyond what the traditional Laplacian is capable of modeling. This, in turn, allows for studying the impact of the interaction between dynamics and network topology on the quality-measure of network clusters and centrality, in order to effectively identify important vertices and communities in the network. Specifically, for each dynamical process in this framework, we define a centrality measure that captures a vertex's participation in the dynamical process on a given network and also define a function that measures the quality of every subset of vertices as a potential cluster (or community) with respect to this process. We show that the subset-quality function generalizes the traditional conductance measure for graph partitioning. We partially justify our choice of the quality function by showing that the classic Cheeger's inequality, which relates the conductance of the best cluster in a network with a spectral quantity of its Laplacian matrix, can be extended to the parameterized Laplacian. The parameterized Laplacian framework brings under the same umbrella a surprising variety of dynamical processes and allows us to systematically compare the different perspectives they create on network structure.

## INTRODUCTION

As flexible representations of complex systems, networks model entities and relations between them as vertices and edges. In a social network for example, vertices are people, and the edges between them represent friendships. As another example, the World Wide Web is a collection of web pages with hyperlinks between them. An unprecedented amount of such relational data is now available. While discovery and fortune await, the challenge is to extract useful information from these large and complex data.

Centrality and community detection are two of the fundamental tasks of network analysis. The goal of centrality identification is to find important vertices that control the dynamical processes taking place on the network. Page Rank (*Page et al., 1999*) is one such

Corresponding author
Kristina Lerman, lerman@isi.edu

measure developed by Google to rank web pages. Other centrality measures, such as degree centrality, Katz score and eigenvector centrality (*Katz, 1953*; *Bonacich, 1972*; *Bonacich & Lloyd, 2001*; *Ghosh & Lerman, 2012*), are used in communication networks for studying how each vertex contributes to the routing of information. Identifying central vertices also plays an important role in methods to maximize influence (*Kempe, Kleinberg & Tardos, 2003*) or limit the spread of a disease on networks.

The objective of community detection is to discover subsets of well-interacting vertices in a given network. Discovering such communities allows us to follow the classic reductionist approach, separating the vertices into distinct classes, each of which can then be analyzed separately. For example, US-based political networks usually exhibit a bipolar structure, representing democrat/republican divisions (*Adamic & Glance, 2005*). Communities within online social networks like Facebook might correspond to real social groups which can be targeted with various advertisements. However, just like with the different notions of centrality, there is an assortment of community detection algorithms, each leading to a different community structure on the same network (see *Fortunato, 2010*; *Porter, Onnela & Mucha, 2009* for reviews).

With so many choices for both centrality and community detection, practitioners often face a difficult decision of which measures to use. Instead of looking for the "best" such measure, we describe an umbrella framework that unifies some of the well known measures, connecting the ideas of centrality, communities and dynamical processes on networks. In this dynamics-oriented view, a vertex's centrality describes its participation in the dynamical process taking place on the network (*Borgatti, 2005*; *Lambiotte et al., 2011*; *Ghosh & Lerman, 2012*). Likewise, communities are groups of vertices that interact more frequently with each other (according to the rules of the dynamical process) than with vertices from other communities (*Lerman & Ghosh, 2012*). In fact, this view of modeling is not new: when choosing *conductance* as a measure of community quality, one implicitly assumes that *unbiased random walk* is taking place on the network (*Kannan, Vempala & Vetta, 2004*; *Spielman & Teng, 2004*; *Chung, 1997*; *Delvenne, Yaliraki & Barahona, 2008*). Under the continuous time random walk model, *heat kernel page rank* (*Chung, 2007*) also leads to a measure of community structures. Other dynamical processes, such as the spread of information, or exchange of opinions, arise from different interactions than the unbiased random walk. For example, maximum entropy random walk (*Burda et al., 2009*) is a stochastic process that is biased towards neighbors that are closer to the network's strongly connected core. Represented by the *replicator* operator (*Lerman & Ghosh, 2012*; *Smith et al., 2013*), it also models an epidemic process at the epidemic threshold, whose stationary distribution is closely related to *eigenvector centrality* (*Bonacich & Lloyd, 2001*; *Ghosh & Lerman, 2011*). It is natural, then, that vertex centrality and community depend on the specifics of the dynamical process, even if the underlying network topologies are the same.

Recently, *Ghosh et al. (2014)* introduced a parameterization of Laplacian operators to capture the interplay between a dynamical process and the underlying topology of the network on which it unfolds. By generalizing the traditional conductance, they proved a more general version of the Cheeger inequality and used it as a basis for an efficient spectral

clustering algorithm (*Spielman & Teng, 2004*; *Andersen, Chung & Lang, 2007*; *Andersen & Peres, 2009*). In this paper, we generalize previous results by introducing a formal framework with additional parameters and better intuitions. We also introduce *parameterized centrality* and relate it to existing centrality measures through transformations. This paper makes the following contributions:

**Parameterized Laplacian ('Parameterized Laplacian Framework'):** We introduce the parameterized Laplacian framework that extends the traditional Laplacian for describing diffusion and random walks on networks. Recall that a random walk is a stochastic dynamical process that transitions from a vertex to a random neighbor of that vertex. It defines a Markov chain that can be specified by the normalized Laplacian of the network. Our framework attempts to capture a family of dynamical processes that have additional parameters based on the normalized Laplacian, which allows the modeling of arbitrary biases and delays. Members of this family are connected via simple parameterized transformations, which enables analysis of the impact of these parameters on the measures of centrality and communities.

**Parameterized centrality ('Parameterized Centrality'):** Based on the connection between centrality measures and the stationary distribution of a random walk (*Page et al., 1999*; *Ghosh & Lerman, 2012*), we generalize the notion of centrality to all dynamical processes in the parameterized Laplacian family. Some well known centrality measures are identified as special cases under this unified framework, which allows us to systematically compare them using transformations. In particular, we show that seemingly different formulations of dynamics are in fact the same after a change of basis. Parameterized centrality also leads to the definition of parameterized volume for subsets of vertices.

**Parameterized conductance ('Parameterized Community Quality'):** We also generalize the notion of conductance to all dynamical processes under the framework and call it parameterized conductance.[1] This quantity measures the quality of every subset (of vertices) as a potential community with respect to this process on the given network. Recall that *conductance* balances between minimizing the cross-community interactions and the volume of each community. Parameterized conductance is defined in exact same fashion, but with the parameterized notions of interaction as well as volume. As with centrality, some existing community measures turn out to be special cases. For completeness, we will restate the previously proven generalized versions of Cheeger inequality and the resulting spectral algorithm (*Ghosh et al., 2014*). The parameterized Laplacian framework enables systematic comparison between different community measures, as they are now unified and connected by simple transformations.

**Empirical evaluation on real-world networks ('Experiments'):** We apply our framework to study the structure of several real-world networks. They are from different domains that embody a variety of dynamical processes and interactions. We contrast the central vertices and communities identified by different dynamical processes and provide an intuitive explanation for their differences. Keep in mind that we do not claim any specific centrality or community structure measures to be the "best." We think every outcome is potentially interesting among many possible perspectives.

[1] A similar measure of a subset of this family is called the generalized conductance in *Ghosh et al., (2014)*.

**Table 1  Glossary of terms and notations.**

| Term | Description | Term | Description |
|---|---|---|---|
| $A$ | Weighted adjacency matrix | $a_{ij}$ | Entry $i,j$ of $A$ |
| $W$ | Interaction matrix | $w_{ij}$ | Entry $i,j$ of $W$ |
| $\boldsymbol{\theta}(t)$ | Vertex state vector (column) at time $t$ | $\theta_i(t)$ | Entry $i$ of $\boldsymbol{\theta}(0)$ |
| $D_A$ | Diagonal degree matrix of $A$ | $d_i$ | Degree of vertex $i$ in $A$ |
| $D_W$ | Diagonal degree matrix of $W$ | $d_{Wi}$ | Degree of vertex $i$ in $W$ |
| $T$ | Diagonal delay matrix | $\tau_i$ | Delay factor of vertex $i$ |
| $\mathcal{L}$ | Generalized Laplacian Operator | $P_{ij}$ | Random walk probability from $j$ to $i$ |
| $\vec{v}_A$ | Dominant eigenvector of $A$ | $\vec{v}_{Ai}$ | Entry $i$ of $\vec{v}_A$ |
| $V_A$ | Diagonal matrix with $\vec{v}_A$ entries | $\vec{v}_i$ | $i$th eigenvector of $\mathcal{L}$ |
| $c_i$ | Centrality of vertex $i$ | $S$ | Subset of $V$, defines a community |

In contrast to the earlier work on which this paper is based, the emphasis of this paper is on the theoretical framework that brings together important concepts in network science. While the parameterized Laplacian framework described in this paper cannot model every dynamical process of interest, it is still flexible enough to include a variety of dynamical processes which are seemingly unrelated. It allows us to systematically study and compare these processes under a unified framework. We hope this study will lead to better approaches for defining and understanding the general interaction between dynamics and topologies.

## BACKGROUND AND RELATED WORK

Before introducing our framework, we briefly review some closely related models. We will later show that these existing models are special cases under the parametrized Laplacian framework. The intuition about these well-known systems is helpful for understanding the motivation behind the framework.

We represent a network as a weighted, undirected graph $G = (V, E, A)$ with $n$ vertices, where for $i, j \in V$, $a_{ij}$ assigns an non-negative weight (affinity) to each edge $(i,j) \in E$. We follow the tradition that $a_{ij} = 0$ if and only if $(i,j) \notin E$; i.e., $A$ is the weighted symmetric adjacency matrix. We assume $a_{ii} = 0$ for all $i \in V$. In the discussion below, the *(weighted) degree* of vertex $i \in V$ is defined as the total weight of edges incident on it, that is, $d_i = \sum_j a_{ij}$. A dynamical process describes a state variable $\theta_i(t)$ associated with each vertex $i$. This variable changes its value based on interactions with the vertex's neighbors according to the rules of the dynamical process.[2]

In this paper, since we view dynamics as operators on the vector composed of vertex state variables, we adopt the linear algebra convention, i.e., using column vertex state vectors $\boldsymbol{\theta}(t)$ and left-multiply them by matrix operators.[3]  Table 1 summarizes the terms and notation.

### Random walks

One of the most-widely studied dynamical processes on networks is the random walk. The simplest is the discrete time *unbiased random walk* (URW), where a walker at vertex

[2]It represents a probability vector in random walks, while becomes a belief vector in consensus processes.

[3]This contrasts with the engineering convention where row vectors and right-multiplications are standards.

**Peer**J Computer Science

[4]When there is no confusion, we will abuse the notation and simply use $\boldsymbol{\theta}$ and $\theta_i$ without their explict time parameter $(t)$.

$i$ follows one of the edges with a probability proportional to the weight of the edge (*Ross, 2014*; *Aldous & Fill, 2002*). In this case, the state vector $\boldsymbol{\theta}^4$ forms a distribution whose expected value follows the update equation:

$$\theta_i(t+1) = \sum_j P_{ij}\theta_j(t).$$

Here $P$ is a stochastic matrix whose entry $P_{ij}$ is the transition probability for a walker to go from the vertex $j$ to $i$, $P_{ij} = a_{ij}/d_j$.

The update equation of an unbiased random walk leads to the difference equation

$$\Delta\theta_i = \theta_i(t+1) - \theta_i(t) = \sum_j P_{ij}\theta_j(t) - \theta_i(t) = -\sum_j L_{ij}^{RW}\theta_j(t),$$

where $L^{RW}$ is the normalized *random walk Laplacian matrix* with $L^{RW} = I - AD_A^{-1}$.

To go from a discrete time synchronous random walk to a continuous time dynamics, we introduce a waiting time function for the asynchronous jumps performed by the walk (*Ross, 2014*). Assuming a simple Poisson process where the waiting times between jumps are exponentially distributed as the PDF $f(t,\tau) = \frac{1}{\tau_i}e^{-\frac{t}{\tau_i}}$, we can rewrite the above difference equations as differential equations,

$$\frac{d\theta_i}{dt} = -\sum_j \frac{L_{ij}^{RW}}{\tau_j}\theta_j.$$

The solution to the above differential equations gives the state vector of the random walk at any time $t$:

$$\boldsymbol{\theta}(t) = e^{-L^{RW}T^{-1}t} \cdot \boldsymbol{\theta}(0),$$

where $T$ is the $n \times n$ diagonal matrix with the mean waiting time $\tau_i$ as entries. If the dynamical process converges, then regardless of its initial value $\boldsymbol{\theta}(0)$, the stationary distribution $\pi_i$ has the following density:

$$\pi_i = \lim_{t\to\infty}\theta_i(t) = \frac{d_i\tau_i}{\sum_j d_j\tau_j}. \tag{1}$$

Intuitively, the stationary distribution is proportional to the product of vertex degree and the mean waiting time.

A natural extension of the process is to bias the random walk towards specific vertices, making it a *biased random walk* (BRW). According to *Lambiotte et al. (2011)*, any biased random walk defined with the transition probability $P_{ij} \propto b_i a_{ij}$ (where $b_i$ is the bias towards vertex $i$) can be reduced to a URW on a re-weighted "interaction network" with the adjacency matrix

$$W = BAB,$$

where $B$ is a diagonal matrix with $B_{ii} = b_i$. The above symmetric re-weighting ensures that

$$P_{ij} = \frac{b_i a_{ij} b_j}{\sum_i b_i a_{ij} b_j} \propto b_i a_{ij}, \qquad P_{ji} = \frac{b_j a_{ji} b_i}{\sum_j b_j a_{ji} b_i} \propto b_j a_{ji}.$$

In one class of BRWs previously studied in network communications (*Ling et al., 2013*; *Fronczak & Fronczak, 2009*; *Gómez-Gardeñes & Latora, 2008*), bias $b_i$ has a power-law dependence on degree: $P_{ij} \propto d_i^\beta a_{ij}$. The exponent $\beta$ controls the strength of bias. The URW is recovered with $\beta = 0$; When $\beta > 0$, biases toward high degree vertices are introduced, and when $\beta < 0$, the random walk is more likely to jump to a lower degree neighbor.

Another type of BRW is the maximum-entropy random walk (*Burda et al., 2009*; *Lambiotte et al., 2011*), defined as

$$\theta_i(t+1) = \sum_j \frac{\vec{v_{A_i}} a_{ij}}{\lambda_{\max} \vec{v_{A_j}}} \theta_j(t),$$

where $\vec{v_A}$ is the eigenvector of $A$ associated with its largest eigenvalue $\lambda_{\max}$: $A\vec{v_A} = \lambda_{\max}\vec{v_A}$. Again, an unbiased random walk on the interaction network $W = V_A A V_A$ is equivalent to biased random walk on the original network $A$ (the entries of diagonal matrix $V_A$ is the components of the eigenvector $\vec{v_A}$). In particular, the stationary distributions of both can be written as $\pi_i = \frac{\vec{v_{A_i}}^2}{\sum_i \vec{v_{A_i}}^2}$.

## Consensus and opinion dynamics

Another closely related class of discrete time dynamical processes is the so-called the "consensus process" (*DeGroot, 1974*; *Lambiotte et al., 2011*; *Olfati-Saber, Fax & Murray, 2007*; *Krause, 2008*). Consensus process models coordination across a network where each vertex updates its "belief" based on the average "beliefs" of its neighbors. Unlike random walks, which conserves total state value throughout the network (since the state vector is always a distribution), the consensus process follows the following update equation

$$\theta_i(t+1) = \frac{1}{d_i} \sum_j a_{ij} \theta_j(t).$$

This leads to the difference equation

$$\Delta \theta_i = \theta_i^{t+1} - \theta_i^t = -\sum_j L_{ij}^{CON} \theta_j(t)$$

where $L^{CON}$ is the *consensus Laplacian matrix* with $L^{CON} = I - D_A^{-1} A$. For an undirected graph with a symmetric $A$, $L^{CON} = [L^{RW}]^T$.

Consensus can also be turned into asynchronous continuous time dynamics. Again, assuming a Poisson process where the update interval at each vertex is exponentially distributed as $\tau_i(t) = \frac{1}{\tau_i} e^{-\frac{t}{\tau_i}}$, we can rewrite the above difference equations as differential equations,

$$\frac{d\theta_i}{dt} = -\sum_j \frac{L_{ij}^{CON}}{\tau_i} \theta_j.$$

The consensus process always converge to a uniform "belief" state with the value,

$$\pi_i = \frac{1}{\sum_j d_j \tau_j} \sum_i \theta_i(0) d_i \tau_i. \tag{2}$$

Just like the URW, unbiased consensus can also be generalized by introducing a weight when averaging over neighbors' values. This opens the door to consensus dynamics such as opinion dynamics (*Krause, 2008*), and linearized approach to synchronization models (*Lerman & Ghosh, 2012*; *Motter, Zhou & Kurths, 2005*; *Arenas, Díaz-Guilera & Pérez-Vicente, 2006*).

## Communities and conductance

In network clustering and community detection, previous work has focused on identifying subsets of vertices $S \subseteq V$ that interact more frequently with vertices in the same community than vertices in other subsets (*Fortunato, 2010*; *Porter, Onnela & Mucha, 2009*). A standard approach to clustering defines an objective function that measures the *quality* of a cluster. For a subset $S \subseteq V$, let $\bar{S} = V \setminus S$ denote the complement of $S$, which consists of vertices that are not in $S$. Let $\text{cut}(S, \bar{S}) = \sum_{i \in S, j \in \bar{S}} a_{i,j}$ denote the total interaction strength of all edges used by $S$ to connect with the outside world. Let $\text{vol}(S) = \sum_{i \in S} d_i = \sum_{i \in S, j \in V} a_{i,j}$ denote the volume of weighted "importance" for all vertices in $S$.

One popular measure of the quality of a subset $S$ as a potential good cluster (or a community) (*Kannan, Vempala & Vetta, 2004*; *Spielman & Teng, 2004*; *Chung, 1997*) is to use the ratio of these two quantities:

$$\phi(S) = \frac{\text{cut}(S, \bar{S})}{\min(\text{vol}(S), \text{vol}(\bar{S}))}. \tag{3}$$

For example, a subset that (approximately) minimizes this quantity—the *conductance* of $S$—is a desirable cluster, as it maximizes the fraction of affinities within the subset. If interactions among vertices are proportional to their affinity weights, then a set with small conductance also means that its members interact significantly more with each other than with outside members. The smallest achievable ratio over all possible subsets is also known as the *isoperimetric number*. As an important measure for mixing time in classic Markov chains, conductance has proven mathematical bounds in terms of the second eigenvalue of its Laplacian (*Cheeger, 1970*; *Jerrum & Sinclair, 1988*; *Lawler & Sokal, 1988*). Other well-known quality functions are normalized cut (*Shi & Malik, 2000*) and ratio-cut, given respectively by

$$\frac{\text{cut}(S, \bar{S})}{\text{vol}(S)} + \frac{\text{cut}(S, \bar{S})}{\text{vol}(\bar{S})} \quad \text{and} \quad \frac{\text{cut}(S, \bar{S})}{\min(|S|, |\bar{S}|)}.$$

Algorithmically, once a quality function is selected, one can then perform a graph partitioning algorithm or any community detection algorithm to find clusters that optimize the objective. The optimization, however, is usually a combinatorial problem. To address this problem on large networks, efficient approximate solutions have been developed,

such as *Spielman & Teng (2004)*, *Andersen, Chung & Lang (2007)*, and *Andersen & Peres (2009)*. Others took a machine learning approach, proposing efficient approximations by enforcing various smoothness and regularization conditions (*Avrachenkov et al., 2011*; *Bertozzi & Flenner, 2012*).

While most community detection algorithms do not explicitly model the dynamical process that defines the interactions between vertices, the connection between conductance and unbiased random walks is quite well studied (*Kannan, Vempala & Vetta, 2004*; *Spielman & Teng, 2004*; *Chung, 1997*). In particular, Chung's work on heat kernel page rank and Cheeger inequality, where a dynamical system is built using the normalized Laplacian, provides a theoretical framework for provably good approximations to the isoperimetric number (*Chung, 2007*). Intuitively, the relationship between clustering and dynamics can be captured as: a community is a cluster of vertices that "trap" a random walk for a long period of time before it jumps to other communities (*Lovász, 1996*; *Shi & Malik, 2000*; *Rosvall & Bergstrom, 2008*; *Spielman & Teng, 2004*). Therefore, the presence of a good cluster based on conductance implies that it will take a random walk a long time to reach its stationary distribution. Similar interplays with community structures can also be generalized to richer dynamical processes, with different time scale, biases and locality settings (*Lambiotte, Delvenne & Barahona, 2008*; *Lambiotte et al., 2011*; *Jeub et al., 2015*).

## PARAMETERIZED LAPLACIAN FRAMEWORK

Consider a linear dynamical process of the following form:

$$\frac{d\boldsymbol{\theta}}{dt} = -\mathcal{L}\boldsymbol{\theta}, \tag{4}$$

where $\boldsymbol{\theta}$ is a column vector of size $n$ containing the values of the dynamical variable for all vertices, and $\mathcal{L}$ is a positive semi-definite matrix, the *spreading operator*, which defines the dynamical process.

As discussed in the introduction, we focus on dynamical processes that generalize the traditional normalized Laplacian for diffusion and random walks. Recall that the *symmetric normalized Laplacian matrix* of a weighted graph $G = (V, E, \boldsymbol{A})$ is defined as $\boldsymbol{D}_{\boldsymbol{A}}^{-1/2}(\boldsymbol{D}_{\boldsymbol{A}} - \boldsymbol{A})\boldsymbol{D}_{\boldsymbol{A}}^{-1/2}$, where $\boldsymbol{D}_{\boldsymbol{A}}$ is the diagonal matrix defined by $(d_1, \ldots, d_n)$. We study the properties of a dynamical process that can be further parameterized as:

$$\mathcal{L}(\rho, \boldsymbol{T}, \boldsymbol{W}) = (\boldsymbol{T}\boldsymbol{D}_{\boldsymbol{W}})^{-1/2-\rho}(\boldsymbol{D}_{\boldsymbol{W}} - \boldsymbol{W})(\boldsymbol{D}_{\boldsymbol{W}}\boldsymbol{T})^{-1/2+\rho}. \tag{5}$$

We name this operator with parameters $\langle \rho, \boldsymbol{T}, \boldsymbol{W} \rangle$ *parameterized Laplacian* and represent it using $\mathcal{L}$ in the rest of the paper. Here $\boldsymbol{T}$ is the $n \times n$ diagonal matrix of *vertex delay factors*. Its $i$th element $\tau_i$ represents the average delay of vertex $i$. We assume that the operator is *properly scaled*: specifically, $\tau_i \geq 1$, for all $i \in V$. Another generalization from the traditional Laplacian is the use of the *interaction matrix* $\boldsymbol{W}$ instead of the adjacency matrix $\boldsymbol{A}$. In theory, $\boldsymbol{W}$ can be any $n \times n$ symmetric positive matrix. Note that the degree matrix $\boldsymbol{D}_{\boldsymbol{W}}$ is now also defined in terms of the interaction matrix, that is $d_{\boldsymbol{W}i} = \sum_j w_{ij}$. While the $\rho$ parameter can technically be any real number, in this work we limit ourselves to three

special cases: $\rho = 1/2, 0, -1/2$. These cases correspond to three equivalent linear operators with "consensus", "symmetric" and "random walk" interpretations respectively.

We show that by transforming the parameterized Laplacian in different ways we can express a number of different dynamic processes. We focus on three simple transformations: (a) the *similarity transformations*, which correspond to the parameter $\rho$ in parameters in Eq. (5), (b) *scaling transformations*, governed by the parameter $T$, and (c) the *reweighing transformation*, governed by $W$.

## Similarity transformations

Changing $\rho$ in Eq. (5) leads to different representations of the same linear operator, unifying seemingly unrelated dynamics, such as "consensus" and "random walk." To see this, we refer to the idea of matrix similarity.

In linear algebra, similarity is an equivalence relation for square matrices. Two $n \times n$ matrices $X$ and $Y$ are similar if

$$X = QYQ^{-1}, \tag{6}$$

where the invertible $n \times n$ matrix $Q$ is called the change of basis matrix. Similar matrices share many key properties, including their rank, determinant and eigenvalues. Eigenvectors are also transforms of each other under a change of basis.

Recall that under our framework, the symmetric version of the parameterized Laplacian matrix is

$$\mathcal{L}^{SYM} = T^{-1/2} D_W^{-1/2} (D_W - W) D_W^{-1/2} T^{-1/2}.$$

We can rewrite the operator describing random walk dynamics as:

$$\mathcal{L}^{RW} = (D_W - W)(D_W T)^{-1} = (D_W T)^{1/2} \mathcal{L}^{SYM} (D_W T)^{-1/2}. \tag{7}$$

Thus, continuous time random walk with delay factors $T$ is similar to the symmetric normalized Laplacian. Similarly, we can rewrite the continuous time consensus dynamics under our framework as

$$\mathcal{L}^{CON} = (D_W T)^{-1}(D_W - W) = (D_W T)^{-1/2} \mathcal{L}^{SYM} (D_W T)^{1/2} = \mathcal{L}^{RW^T}. \tag{8}$$

The fact that "consensus," "symmetric" and "random walk" operators are similar means that they model the same dynamics on a network, provided that we observe them in a consistent basis.

The random walk Laplacian matrix provides a physical intuition for our framework. An unbiased random walk on the interaction graph $W$ is equivalent to a biased random walk on the original adjacency matrix $A$ (*Lambiotte et al., 2011*). On the other hand, $\tau_i$ specifies the mean delay time of the random walk on vertex $i$ before a transition. This interpretation reveals the orthogonal nature of the parameters: namely $W$ controls the distribution of walk trajectories while $T$ controls the delay time of vertex transitions along each trajectory.

While we use symmetric operators for mathematical convenience in definitions and proofs and abuse the notation $\mathcal{L} = \mathcal{L}^{SYM}$, it is often more intuitive to think from the

random walk or consensus perspective. In the following subsections, we will use the random walk formulation ($\rho = -1/2$) as examples, but all results apply to arbitrary $\rho$ values under a simple change of basis. More discussion about the similarity transformation follows after we introduce a few properties of the parameterized Laplacian.

## Scaling transformations

*Uniform scaling.*  One of the simplest transformations is uniform scaling, which is given by the diagonal matrix $T$ with identical entries:

$$X = YQ = \gamma Y, \tag{9}$$

where the scalar matrix $Q$ can be rewritten as $\gamma I$, where $\gamma$ is a scalar. Uniform scaling preserves almost all matrix properties, including the eigenvalue and eigenvector pairs associated with the operator.

Intuitively, uniform scaling can be understood as rescaling time by $1/\gamma$. In other words, a bigger global "time delay" slows down the random walk. Uniform scaling is a useful transformation that enables the parameterized Laplacian to include arbitrary time delay factors $T'$. The trick is to rescale $T$ to meet the condition $\tau_i \geq 1$ by making $T = \frac{T'}{\max_i \tau_i}$ without affecting any other matrix properties. We will later use it to define special operators under the framework.

*Non-uniform scaling.*  Non-uniform scaling enables us to use the $T$ parameter to control the time delay at each vertex. Non-uniform scaling is written as

$$X = YQ, \tag{10}$$

where the diagonal matrix $Q$ can have different entries. Unlike uniform scaling, this scaling does not preserve the matrix's spectral properties.

Under the parameterized Laplacian framework, non-uniform scaling can be understood as rescaling the mean waiting time at each vertex $i$ by $\tau_i$. Non-uniform scaling does not affect the trajectory of the random walk: the sequence of vertices, $i_0, i_1, \ldots, i_t$, visited by the random walk during some time interval does not depend on $T$. What changes with $T$ is only the waiting time at each vertex, i.e., the time the walk spends on the vertex before a transition.

## Reweighing transformations

The last parameterization we explore is one that transforms the adjacency matrix of a graph, $A$, to the interaction matrix $W$. Given an adjacency matrix $A$, the choice of $W$ is a rather flexible design option. In fact, we can arbitrarily manipulate the adjacency matrix as long as the result is still a positive and symmetric matrix, for any perceived dynamics.

In this paper, we limit our attention to bias transformations of the original adjacency matrix $A$. We call them the *reweighing transformations*. Whereas the scaling transformation changes the delay time at each vertex, the reweighing transformation changes the trajectory of the dynamic process. Note that this transformation also changes the degree matrix $D_W$.

As described in 'Background and Related Work', a biased random walk with transition probability $P_{ij} \propto b_i a_{ij}$ is equivalent to an unbiased random walk on an "interaction graph," represented by the reweighed adjacency matrix:

$$w_{ij} = b_i a_{ij} b_j, \tag{11}$$

where we constrain $b_i > 0$. This transformation allows the parameterized Laplacian to model many different types of dynamic processes by transforming them into a unbiased walk on the reweighted interaction graph.

## Special cases

The simple parameterization of the Laplacian in terms of $T$ and $W$ allows us to model a variety of dynamic processes,[5] including those described by the Laplacian and normalized Laplacian, as well as a continuous family of new operators that are not as well studied. It also contains operators for modeling some types of epidemic processes. The consideration of this family of operators is also partially motivated by recent experimental work in understanding network centrality (*Ghosh & Lerman, 2011*; *Lerman & Ghosh, 2012*).

*Normalized Laplacian.* If the interaction matrix is the original adjacency matrix of the graph $W = A$, and vertex delay factor is simply the identity matrix $T = I$, then we recover the *symmetric normalized Laplacian*:

$$\mathcal{L} = I - D_A^{-1/2} A D_A^{-1/2}.$$

The "random walk" and "consensus" formulations of this dynamic process correspond to the unbiased random walk and consensus processes described in 'Background and Related Work': $\mathcal{L}^{RW} = I - A D_A^{-1}$ and $\mathcal{L}^{CON} = I - D_A^{-1} A$.

*(Scaled) Graph Laplacian.* When $W = A$, $T = d_{\max} D_A^{-1}$, the parameterized Laplacian operator corresponds to the (scaled) graph *Laplacian*

$$\mathcal{L} = \frac{1}{d_{\max}} (D_A - A).$$

This operator is often used to describe *heat diffusion* processes (*Chung, 2007*), where $\mathcal{L}$ is replacing the continuous Laplacian operator $\nabla^2$.

Notice that by setting $T = d_{\max} D_A^{-1}$, the diagonal matrix $T D_W$ becomes effectively a scalar. As a result, different similarity transformation (other values of $\rho$ in Eq. (5)) lead to identical linear operators, meaning the "random walk" and "consensus" formulations are exactly the same as the symmetric formulation.

*Replicator.* Let $\overrightarrow{v_A}$ be the eigenvector of $A$ associated with its largest eigenvalue $\lambda_{\max}$: $A\overrightarrow{v_A} = \lambda_{\max} \overrightarrow{v_A}$. We can then construct a diagonal matrix $V_A$ whose elements are the components of the eigenvector $\overrightarrow{v_A}$. Let us scale the adjacency matrix according to $W = V_A A V_A$ and use it as the interaction matrix. Setting the vertex delay factor to identity, the spreading operator is:

$$\mathcal{L} = I - D_W^{-1/2} W D_W^{-1/2} = I - \frac{1}{\lambda_{\max}} A,$$

[5] By fixing $\rho = 0$ we recover the family of special cases considered in *Ghosh et al. (2014)*.

where the entries in $\boldsymbol{D_W}$ simplifies as $d_{Wi} = \sum_j \vec{v_{A_i}} a_{ij} \vec{v_{A_j}} = \vec{v_{A_i}} \sum_j a_{ij} \vec{v_{A_j}} = \lambda_{\max} \vec{v_{A_i}}^2$. This operator is known as the replicator matrix $\boldsymbol{R}$, and it models *epidemic diffusion at the epidemic threshold* on a graph (*Lerman & Ghosh, 2012*). It is simply the normalized Laplacian of the interaction graph $\boldsymbol{V_A A V_A}$ (*Smith et al., 2013*), given by reweighting the adjacency graph $\boldsymbol{A}$ with the eigenvector centralities of the vertices.

Using the random walk formulation, an URW on $\boldsymbol{V_A A V_A}$ is equivalent to a maximum entropy random walk on the original graph $\boldsymbol{A}$ (*Burda et al., 2009*; *Lambiotte et al., 2011*). Its solution is

$$\theta_i(t+1) = \sum_j \frac{\vec{v_{A_i}} w_{ij}}{\lambda_{\max} \vec{v_{A_j}}} \theta_j(t). \tag{12}$$

This means that both dynamics have exactly the same state vector $\theta$ at each time step. In particular, the stationary distributions are both $\pi_i = \frac{\vec{v_{A_i}}^2}{\sum_i \vec{v_{A_i}}^2}$.

The consensus formulation of the replicator gives a maximum entropy agreement dynamics:

$$\mathcal{L}^{CON} = \boldsymbol{I} - \frac{1}{\lambda_{\max}} \boldsymbol{V_A}^{-1} \boldsymbol{A} \boldsymbol{V_A}.$$

*Unbiased Laplacian.* Reweighing each edge by the inverse of the square root of the endpoint degrees gives the what is known as the normalized adjacency matrix $\boldsymbol{W} = \boldsymbol{D_A}^{-1/2} \boldsymbol{A} \boldsymbol{D_A}^{-1/2}$ (*Chung, 1997*). Then, the degree of vertex $i$ of the reweighted graph is $d_{Wi} = \sum_{j \in V} \boldsymbol{W}[i,j]$. With $\boldsymbol{T} = d_{W\max} \boldsymbol{D_W}^{-1}$ we define the *unbiased Laplacian matrix*:

$$\mathcal{L} = \frac{1}{d_{W\max}} (\boldsymbol{D_W} - \boldsymbol{W}).$$

Unbiased Laplacian is an example of the degree based biased random walk with $P_{ij} \propto d_i^{-1/2} a_{ij}$ ('Background and Related Work'). An URW on the reweighed adjacency matrix $\boldsymbol{W}$ is equivalent to a BRW on the original adjacency matrix of the following dynamics

$$\theta_i(t+1) = \sum_j \frac{d_i^{-1/2} a_{ij}}{\sum_k d_k^{-1/2} a_{kj}} \theta_j(t). \tag{13}$$

The stationary distribution for this class of BRWs in general is $\pi_i = \frac{\sum_i d_i^\beta a_{ij} d_j^\beta}{\sum_{ij} d_i^\beta a_{ij} d_j^\beta}$.

Equivalent to the (scaled) graph Laplacian of the normalized adjacency matrix, the diagonal matrix $\boldsymbol{T D_W}$ of the unbiased Laplacian is also effectively a scalar. As a result, the "random walk" and "consensus" formulations are exactly the same as the symmetric formulation.

These four special cases are related to each other through various transformations introduced earlier in this section, which are captured by Fig. 1.

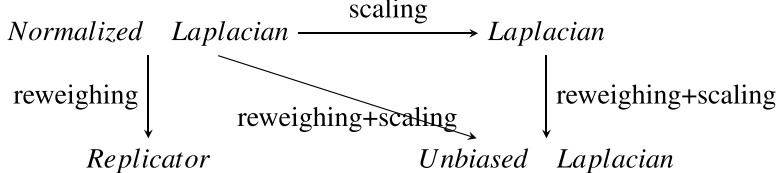

**Figure 1** Relationships between special cases of the parameterized Laplacian.

# PARAMETERIZED CENTRALITY

Centrality is used to capture how "central" or important a vertex is in a network. In dynamical systems, a centrality measure should have the following properties: (1) it should be a per-vertex measure, with all values positive scalars; (2) it should be strongly related to that's vertex's state variable; (3) it should be independent of initial state of the state vector. These conditions ensure that centrality of a vertex is determined by the topology of the network as well as the interactions taking place on it. It also follows our intuition that the importance of a vertex should not depend on the specific initializations of the dynamical process. It is sometimes desirable to define a centrality measure as a function of time (*Taylor et al., 2015*). In this paper, however, we stick to the more conventional notion of time-invariant centralities.

The various centrality measures introduced in the past have lead to very different conclusions about the relative importance of vertices (*Katz, 1953*; *Bonacich, 1972*; *Page et al., 1999*), including degree centrality, eigenvector centrality and PageRank. Our parameterized Laplacian framework unifies some of these measures by showing that they are related to solutions of different dynamic processes on the network.

## Stationary distribution of a random walk

A vertex has high centrality with respect to a random walk if it is visited frequently by it. This is specified by the distribution of the dynamic process at time $t$:

$$\boldsymbol{\theta}(t) = e^{-\boldsymbol{\mathcal{L}}^{RW}t} \cdot \boldsymbol{\theta}(0) = \sum_{k=0}^{\infty} \frac{(-t)^k}{k!} \boldsymbol{\mathcal{L}}^{RW^k} \boldsymbol{\theta}(0), \tag{14}$$

where $\boldsymbol{\theta}(0)$ is the state vector describing the initial distribution of the random walk. The stationary distribution of the random walk:

$$\lim_{t \to \infty} \boldsymbol{\theta}(t) = \boldsymbol{\pi} \quad \text{with} \quad \pi_i = \frac{d_{Wi}\tau_i}{\sum_j d_{Wj}\tau_j}, \tag{15}$$

because

$$(\boldsymbol{D_W} - \boldsymbol{W})(\boldsymbol{TD_W})^{-1}\Pi = (\boldsymbol{D_W} - \boldsymbol{W})\overrightarrow{1} = \overrightarrow{0},$$

with $\boldsymbol{\pi}$ being the vector with $\pi$ entries and $\Pi$ being the diagonal matrix with the same elements. By convention, $\boldsymbol{\pi}$ is the standard centrality measure in conservative processes, including random walks (*Ghosh & Lerman, 2012*).

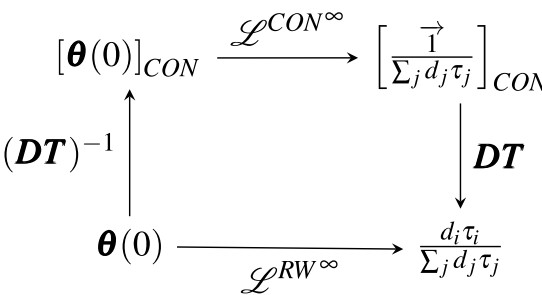

**Figure 2** The similarity transformations between "consensus" and "random walk" dynamics.

If we define centrality as the stationary distribution of a random walk, the importance of a vertex can be thought of as the total time a random walk spends at the vertex in the steady state. This is proportional to both vertex degree and delay factor, which we will later relate to the volume measure. If $\mathcal{L}^{RW}$ is a normalized Laplacian, this centrality measure is exactly the heat kernel page rank (*Chung, 2009*), which is identical to degree centralities since $W = A$ and $T = I$.

## Stationary distribution of consensus dynamics

In consensus processes, the state vector always converges to a uniform state, where each vertex has the same value of the dynamic variable. As a result, the stationary distribution is not an appropriate measure of vertex centrality, since it deem all vertices to be equally important. However, the final consensus value associated with each vertex is

$$\pi_i = \frac{1}{\sum_j d_{Wj}\tau_j} \sum_{i \in V} \theta_i(0) d_{Wi}\tau_i, \tag{16}$$

where weight of vertex $i$ in this average is $\frac{d_{Wi}\tau_i}{\sum_j d_{Wj}\tau_j}$.

Intuitively, as a measure of importance, it make sense to define the centrality of a vertex in the consensus process as its contribution to the final value. This consistency between "consensus" and "random walk" leads us to define the parameterized centrality.

## Parameterized centrality

As shown in 'Similarity transformations,' the matrices connected through a similarity transformation represent the same linear operator up to a change of basis. For example, the relationship between "consensus" and "random walk" dynamics are captured by Fig. 2.

The above equivalence applies to all state vectors at any time $t$, including the stationary state. To verify, we first rewrite the initial state vector in terms of the eigenvectors of $\mathcal{L}$ $\{\vec{v_1}, \vec{v_2}, \ldots, \vec{v_n}\}$, indexed by their corresponding eigenvalues in ascending order $\lambda_1 < \lambda_2 < \cdots$, with the smallest $\lambda_1$ as the dominant eigenvalue.

**Table 2 Stationary and initial state vectors of different formulations of the parameterized Laplacian.**

| Formulations | $[\theta(0)]_\rho$ | $u_{1i}$ | $z_1$ | $\vec{v}_{1i}$ | $[\pi_i]_\rho$ |
|---|---|---|---|---|---|
| $\mathcal{L}^{SYM}$ | $(DT)^{-1/2}\theta(0)$ | $\frac{\sqrt{d_i\tau_i}}{\sqrt{\sum_j d_j\tau_j}}$ | $\frac{1}{\sqrt{\sum_j d_j\tau_j}}$ | $\frac{\sqrt{d_i\tau_i}}{\sqrt{\sum_j d_j\tau_j}}$ | $\frac{\sqrt{d_j\tau_j}}{\sum_j d_j\tau_j}$ |
| $\mathcal{L}^{RW}$ | $\theta(0)$ | $\frac{1}{\sqrt{\sum_j d_j\tau_j}}$ | $\frac{1}{\sqrt{\sum_j d_j\tau_j}}$ | $\frac{d_i\tau_i}{\sqrt{\sum_j d_j\tau_j}}$ | $\frac{d_j\tau_j}{\sum_j d_j\tau_j}$ |
| $\mathcal{L}^{CON}$ | $(DT)^{-1}\theta(0)$ | $\frac{d_i\tau_i}{\sqrt{\sum_j d_j\tau_j}}$ | $\frac{1}{\sqrt{\sum_j d_j\tau_j}}$ | $\frac{1}{\sqrt{\sum_j d_j\tau_j}}$ | $\frac{1}{\sum_j d_j\tau_j}$ |

$$\theta(t) = e^{-\mathcal{L}t} \cdot \theta(0) = \sum_{k=0}^{\infty} \frac{(-t)^k}{k!} \mathcal{L}^k \theta(0)$$

$$= \sum_i \sum_{k=0}^{\infty} \frac{(-t)^k}{k!} \lambda_i{}^k z_i \vec{v_i} = \sum_i z_i e^{-\lambda_i t} \vec{v_i}$$

$$= \sum_i \boldsymbol{u}_i^T \theta(0) e^{-\lambda_i t} \vec{v_i} = \mathbb{V} e^{-\Lambda t} \mathbb{U}^T \theta(0), \tag{17}$$

where in the last step we used matrices to simplify the notation, with $\Lambda$ being the diagonal matrix of eigenvalues, $\mathbb{V}$ composed of $\{\vec{v_1}, \vec{v_2}, \ldots, \vec{v_n}\}$ as columns and $\mathbb{U}^T = \mathbb{V}^{-1}$. One interesting observation is that by left multiplying both sides with $\mathbb{U}^T$, we have

$$\mathbb{U}^T \theta(t) = \mathbb{U}^T \mathbb{V} e^{-\Lambda t} \mathbb{U}^T \theta(0) = e^{-\Lambda t} \mathbb{U}^T \theta(0).$$

Recall that $\mathbb{U}^T \theta$ is a vector in the eigenbasis $\mathbb{V}$. Applying the operator $\mathcal{L}$ to any input vector simply re-scales it according to eigenvalues. Since the smallest eigenvalue of the parameterized Laplacian is always 0, we have

$$\boldsymbol{u}_1^T \theta(t) = e^{-\lambda_1 t} \boldsymbol{u}_1^T \theta(0) = \boldsymbol{u}_1^T \theta(0),$$

which states that the state vector is conserved along the direction of the dominant eigenvector $\vec{v_1}$.

The state vector reaches a stationary distribution $\pi$

$$\pi = \lim_{t \to \infty} \theta(t) = \lim_{t \to \infty} e^{\lambda_1 t} \theta(t)$$

$$= z_1 \left(\frac{e^{\lambda_1}}{e^{\lambda_1}}\right)^t \vec{v_1} + z_2 \left(\frac{e^{\lambda_1}}{e^{\lambda_2}}\right)^t \vec{v_2} + \cdots + z_n \left(\frac{e^{\lambda_1}}{e^{\lambda_n}}\right)^t \vec{v_n} \approx z_1 \vec{v_1}. \tag{18}$$

Since all terms vanish as $t \to \infty$, the stationary state vector $\pi$ only depends on $\vec{v_1}$. $z_1 \vec{v_1}$ qualifies as a time invariant, initialization-independent vertex centrality measure.

Table 2 summarizes the properties of the stationary distributions and centralities associated with different similarity transformation of the parameterized Laplacian. $[\theta]_\rho$ represents the vector $\theta$ under the basis specified by the $\rho$ parameter, with the random walk vector under the standard basis being $\theta(0)$.

The spectral theorem states that any symmetric real matrix, regardless if its rank, has an orthonormal basis $\mathbb{V}$ which consists of its eigenvectors. Under the parameterized Laplacian

framework, the symmetric formulation with $\rho = 0$ falls into this category. In the above table, we have chosen the normalization of the orthonormal basis $\sqrt{\sum_j d_j \tau_j}$ as the common normalization for all formulations.

As the table shows, similarity transformations of the same operator give the same the state vector $\boldsymbol{\theta}$, as long as the input and output vectors are properly transformed into the correct basis. They represent the same dynamics in different coordinate systems. Since centrality is determined by the dynamic process on a given network, it should be unified across these similarity transformations. In theory, any coordinate system can be set as the standard. Here, following the intuitions described earlier, we define the unnormalized stationary state vector of the random walk as the *parameterized centrality*:

$$c_i = d_{W\,i}\tau_i. \tag{19}$$

Another motivation behind this definition is to establish a direct connection between centrality and community measures, as we will later demonstrate with the notion of *parameterized volume* (23).

### Transformations and special cases

Parameterized centrality includes many well known centrality measures as special cases. Below, we summarize the induced special cases discussed in the previous subsection.

*Normalized Laplacian.* $\boldsymbol{W} = \boldsymbol{A}$ and $\boldsymbol{T} = \boldsymbol{I}$, and hence the parameterized centrality reduces to degree centrality $c_i = d_i$.

*(Scaled) Graph Laplacian.* $\boldsymbol{W} = \boldsymbol{A}$ and $\boldsymbol{T} = d_{\max}\boldsymbol{D}_A^{-1}$, hence the parameterized centrality measure here is uniform with $c_i = d_{\max}$. This intuition is easier to see if one considers the unnormalized Laplacian as a consensus operator, as it is often used to calculate the unweighted average of vertex states (*Olfati-Saber, Fax & Murray, 2007*).

*Replicator.* $\boldsymbol{W} = \boldsymbol{V}_A \boldsymbol{A} \boldsymbol{V}_A$ and $\boldsymbol{T} = \boldsymbol{I}$. Recall that $\overrightarrow{v_A}$ is the eigenvector of $\boldsymbol{A}$ associated with the largest eigenvalue $\lambda_{\max}$. The parameterized centrality in this case is $c_i = \lambda_{\max}\overrightarrow{v_{A_i}}^2$, which corresponds to the stationary distribution of a maximal-entropy random walk on the original graph $\boldsymbol{A}$. Note that $\overrightarrow{v_A}$, also known as the *eigenvector centrality*, was introduced by Bonacich (*Bonacich & Lloyd, 2001*) to explain the importance of actors in a social network based on the importance of the actors to which they were connected.

*Unbiased Laplacian.* $\boldsymbol{W} = \boldsymbol{D}_A^{-1/2}\boldsymbol{A}\boldsymbol{D}_A^{-1/2}$ and $\boldsymbol{T} = d_{W\max}\boldsymbol{D}_W^{-1}$. Similar to the (scaled) graph Laplacian, the parameterized centrality measure here is uniform with $c_i = d_{W\max}$.

*Other transformations.* Besides the above special cases, we can use any transformation introduced in the last section for new dynamics, and the corresponding parameterized centrality will be immediately apparent. Scaling transformations change $\tau_i$ terms, while reweighing transformations change $d_{W\,i}$. Similarity transform has no effect on parameterized centrality by definition.

## PARAMETERIZED COMMUNITY QUALITY

Now we investigate the impact of dynamics on network communities. A community is a subset of vertices that interact more with each other according to the rules of a dynamic process than with outside vertices. A *quality function* measures the degree to which this interaction is confined within communities. Here in the context of dynamical processes, we use the following considerations to constrain our choice of quality function: 1. Community quality should be a global measure of interactions; 2. Community quality should be invariant of initial state vectors; 3. Community quality of a subset should be strongly correlated to the change of state variable of member vertices.

The above conditions ensure that the quality function is solely determined by the choice of communities, network structure and the interactions between vertices. We assume that the underlying network structure remains static as the dynamics unfolds. Similar to parameterized centralities, we focus on the time-invariant communities. There is a catch, however, by simply dividing each vertex into its own community, we would have a optimal but trivial community division. Therefore, we need additional constraint on the size of the communities.

A closely related problem in geometry is the isoperimetric problem, which relates the circumference of a region to its area. Isoperimetric inequalities lie at the heart of the study of expander graphs in graph theory. In graphs, area translates into the size of the vertex subset, and the circumference translates into the size of their boundary (*Chung, 1997*). In particular, we will focus on the graph bisection (cut) problem, which restricts the number of communities to two. For bisections, the constraint on community sizes becomes a balancing problem.

Just as for centrality, various community measures used in previous literature lead to very different conclusions about community structure (*Fortunato, 2010*; *Newman, 2006*; *Rosvall & Bergstrom, 2008*; *Zhu, Yan & Moore, 2014*). In this section, we will demonstrate that for graph bisection, some of them are essentially graph isoperimetric solutions under our parameterized Laplacian framework, and more importantly, each one corresponds to a unified community measure for a class of similar operators including seemingly different formulations of "consensus," "symmetric" and "random walk."[6]

### Parameterized conductance

Recall that conductance is a community quality measure associated with unbiased random walks.

$$\phi(S) = \frac{\text{cut}_A(S, \bar{S})}{\min(\text{vol}(S), \text{vol}(\bar{S}))}, \tag{20}$$

where $\text{vol}(S) = \sum_{i \in S} d_i$ and $\text{cut}_A = \sum_{i \in S, j \in \bar{S}} a_{ij}$.

We generalize this notion with a claim that every dynamic process has an associated function that measures the quality of the cluster with respect to that process. Optimizing the quality function leads to cohesive communities, i.e., groups of vertices that "trap" the specific dynamic process for a long period of time.

[6]We focus on the symmetric formulation in this section for mathematical convenience, but all results apply for general $\rho$ values under similarity transformations.

Consider a dynamic process defined by the spreading operator $\mathcal{L} = T^{-1/2}D_W^{-1/2}(D_W - W)D_W^{-1/2}T^{-1/2}$. We define the *parameterized conductance* of a set $S$ with respect to $\mathcal{L}$ as:

$$h_{\mathcal{L}}(S) = \frac{\text{cut}_W(S, \bar{S})}{\min\left(\text{vol}_{\mathcal{L}}(S), \text{vol}_{\mathcal{L}}(\bar{S})\right)} \qquad (21)$$

$$= \frac{\displaystyle\sum_{i \in S, j \in \bar{S}} w_{ij}}{\min\left(\displaystyle\sum_{i \in S} d_{W i}\tau_i, \sum_{i \in \bar{S}} d_{W i}\tau_i\right)}. \qquad (21)$$

The minimum over all possible $S$ is the parameterized conductance of the graph,

$$\phi_{\mathcal{L}}(G) = \min_{S \in V} h_{\mathcal{L}}(S). \qquad (22)$$

Notice that we have also defined the *parameterized volume* of a set $S \subseteq V$ as

$$\text{vol}_{\mathcal{L}}(S) = \sum_{i \in S} c_i = \sum_{i \in S} d_{W i}\tau_i, \qquad (23)$$

which is the sum of parameterized centralities of member vertices. Using the random walk perspective, the numerator measures the random jumps across communities, while the denominator ensures a balanced bisection. As previously pointed out, the presence of a good cut implies that it will take a random walk a long time to cross this boundary and reach its stationary distribution. This corresponds to a small numerator. The parameterized volume can be interpreted as the total time a random walk stays within a community after convergence, as it is proportional to both vertex degrees and vertex delay factors. This interpretation of the denominator coincides with our definition of parameterized centrality (19).

## Transformations and special cases

We can use any transformation to produce new dynamics, and the corresponding parameterized conductance will be redefined according to Eq. (21),

$$h_{\mathcal{L}}(S) = \frac{\displaystyle\sum_{i \in S, j \in \bar{S}} w_{ij}}{\min\left(\displaystyle\sum_{i \in S} d_{W i}\tau_i, \sum_{i \in \bar{S}} d_{W i}\tau_i\right)}. \qquad (24)$$

However, the effect of transformations on the resulting communities is not as obvious when compared with the parameterized centrality. Below, we elaborate the effect of transformations on the parameterized conductance measure in cases and examples.

First of all, the similarity transformation keeps both numerator and denominator the same, which makes the quality function of the same communities identical. This ultimately leads to identical parameterized conductances, which is the minimum over all possible

bisections. Uniform scaling does change the denominator. However, because all possible bisections are scaled uniformly, the relative quality measure remain the same, leading to identical parameterized conductances communities.

From the algorithmic perspective, both similarity and uniform scaling transformations preserve spectral properties of the operator. Since the spectrum is the only input information our spectral dynamics clustering Algorithm 1 uses, we always expect to get the same solution after the transformations. This is not the case with non-uniform scaling and reweighing transformations.

With non-uniform scaling, the numerator remains unchanged. It is each vertex's delay time change that scales the volume measures in the denominator, which in turn results in different optimal bisections because of the balance constraint.

The reweighing transformation is the most complex of all, changing both the numerator and denominator in Eq. (21). This trade-off between cut and balance can oftentimes be very complicated to analyze (as will be seen with real world networks).

Finally, we summarize the induced special cases.

*Normalized Laplacian.* $W = A$ and $T = I$, and hence $h_{\mathcal{L}}(S)$ is the conductance.

*(Scaled) Graph Laplacian.* $W = A$ and $T = d_{\max}D_A^{-1}$, hence

$$h_{\mathcal{L}}(S) = \frac{\mathrm{cut}_A(S,\bar{S})}{\min(d_{\max}|S|, d_{\max}|\bar{S}|)} = \frac{1}{d_{\max}} \cdot \frac{\sum\limits_{i \in S, j \in \bar{S}} a_{ij}}{\min(|S|, |\bar{S}|)}.$$

This is the ratio cut scaled by $1/d_{\max}$.

*Replicator.* $W = V_A A V_A$ and $T = I$. Recall $\overrightarrow{v_A}$ is the eigenvector of $A$ associated with the largest eigenvalue $\lambda_{\max}$. The redefined cut size is $\sum_{i \in S, j \in \bar{S}} w_{ij} = \sum_{i \in S, j \in \bar{S}} \overrightarrow{v_{A_i}} a_{ij} \overrightarrow{v_{A_j}}$. Therefore,

$$h_{\mathcal{L}}(S) = \frac{\sum\limits_{i \in S, j \in \bar{S}} \overrightarrow{v_{A_i}} a_{ij} \overrightarrow{v_{A_j}}}{\lambda_{\max} \min\left(\sum\limits_{i \in S} \overrightarrow{v_{A_i}}^2, \sum\limits_{i \in \bar{S}} \overrightarrow{v_{A_i}}^2\right)}.$$

Since the degree of a vertex in the interaction graph $W$ is $d_{W_i} = \sum_j w_{ij} = \lambda_{\max} \overrightarrow{v_{A_i}}^2$, the parameterized conductance of the replicator is simply the conductance of the interaction graph $W$ (*Smith et al., 2013*).

*Unbiased Laplaican.* $W = D_A^{-1/2} A D_A^{-1/2}$ and $T = d_{W\max} D_W^{-1}$. The associated quality function is

$$h_{\mathcal{L}}(S) = \frac{1}{d_{W\max}} \cdot \frac{\sum\limits_{i \in S, j \in \bar{S}} \frac{a_{ij}}{\sqrt{d_i d_j}}}{\min(|S|, |\bar{S}|)}.$$

Notice that here the parameterized conductance for graph Laplacian and unbiased Laplacian share the same denominator even though they are related through both reweighing and scaling transformations. This is a result of their scaling cancelling out the reweighing effect on volumes (centralities). This is part of the motivation behind our design of the unbiased Laplacian operator for easier comparisons. Another simple obseravation is that graph Laplacian shares the same numerator with its normalized counterpart. We will be using these relationships for analyzing experimental results in the next section.

## Parameterized Cheeger inequality

Given the parameterized conductance measure, finding the best community bisection is still a combinatorial problem, which quickly becomes computationally intractable as the network grows in size. In this subsection we will extend the theorems for the classic Laplacian to our parameterized setting, ultimately leading to efficient approximate algorithms with theoretical guarantees. For mathematical convenience we will use the symmetric formulation and assume that $\rho = 0$ for $\mathcal{L}$. Cheeger inequality (*Cheeger, 1970*) states that

$$\phi^2(G)/2 \leq \lambda_2 \leq 2\phi(G)$$

where $\lambda_2$ is the second smallest eigenvalue of the symmetric normalized Laplacian, $\mathcal{L} = I - D^{-1/2}WD^{-1/2}$, and $\phi(G)$ is conductance. The relationship between conductance and spectral properties of the Laplacian enables the use of its eigenvectors for partitioning graphs, particularly the nearest-neighbor graphs and finite-element meshes (*Spielman & Teng, 1996*).

In this section, we generalize Cheeger inequality to any spreading operator under our framework and its associated parameterized conductance of the graph (given by Eq. (22)). Compared with classic results in Markov chain mixing times (*Jerrum & Sinclair, 1988*; *Lawler & Sokal, 1988*), we generalize Cheeger inequality to accommodate the asycronized delay factors in $T$. It also comes with algorithmic consequences, leading to spectral partitioning algorithms that are efficient in finding low conductance cuts for a given operator.

**Theorem 1. (Parameterized Cheeger Inequality):** *Consider the dynamic process described by a (properly scaled) spreading operator $\mathcal{L} = T^{-1/2}D_W^{-1/2}(D_W - W)D_W^{-1/2}T^{-1/2}$. Let $\lambda_1 \leq \lambda_2 \leq \cdots \leq \lambda_n$ be the eigenvalues of $\mathcal{L}$. Then $\lambda_1 = 0$ and $\lambda_2$ satisfies the following inequalities:*

$$\phi_{\mathcal{L}}(G)^2/2 \leq \lambda_2 \leq 2\phi_{\mathcal{L}}(G)$$

*where $\phi_{\mathcal{L}}(G)$ is given by Eq. (22).*

**Proof.** We prove the theorem by following the approach for proving the classic Cheeger inequality (see *Chung, 1997*).

Let $(\tau_1, \ldots, \tau_n)$ be the diagonal entries of $T$, and $\overrightarrow{v_1}$ be the eigenvector associated with $\lambda_1$. Note that $\overrightarrow{v_1} = T^{1/2} D_W^{1/2} \cdot \overrightarrow{1}$, where $\overrightarrow{1}$ denotes the column vector of all 1's, is an eigenvector of $\mathcal{L}$ associated with eigenvalue $\lambda_0 = 0$. Let $\text{vol}_{\mathcal{L}}(S) = \sum_{i \in S} d_i \tau_i$ for $S \subseteq V$, where for clarity we abuse the notation $d_i$ and use it as $d_{Wi}$.[7] Suppose $f$ is the eigenvector associated with $\lambda_2$. Then, $f \perp \overrightarrow{v_1}$. Consider vector $g$ such that $g[u] = f[u]/\sqrt{d_u \tau_u}$. The fact that $f \perp \overrightarrow{v_1}$ then implies $\sum_v g[v] d_v \tau_v = 0$. Then,

[7] We shall revert back to $d_{Wi}$ notations after this proof.

$$\lambda_2 = \frac{f^T \mathcal{L} f}{f^T f} = \frac{\displaystyle\sum_{u,v \in V} \left( \frac{f[u]}{\sqrt{d_u \tau_u}} - \frac{f[v]}{\sqrt{d_v \tau_v}} \right)^2 w_{u,v}}{\displaystyle\sum_v f[v]^2}$$

$$= \frac{\displaystyle\sum_{u,v \in V} \left( g[u] - g[v] \right)^2 w_{u,v}}{\displaystyle\sum_v g[v]^2 d_v \tau_v}.$$

Instead of sweeping the vertices of $G$ according to the eigenvector $f$ itself, we sweep the vertices of the graph $G$ according to $g$ by ordering the vertices of $G$ so that

$$g[v_1] \geq g[v_2] \geq \cdots \geq g[v_n]$$

and consider sets $S_i = \{v_1, \ldots, v_i\}$ for all $1 \leq i \leq n$.

Similar to *Chung (1997)*, we will eventually only consider the first "half" of the sets $S_i$ during the sweeping: let $r$ denote the largest integer such that $\text{vol}_{\mathcal{L}}(S_r) \leq \text{vol}_{\mathcal{L}}(V)/2$. Note that

$$\sum_v (g[v] - g[v_r])^2 d_v \tau_v = \sum_v g[v]^2 d_v \tau_v + g[v_r]^2 d_v \tau_v \geq \sum_v g[v]^2 d_v \tau_v$$

where the first equation follows from $\sum_v g[v] d_v \tau_v = 0$. We denote the positive and negative part of $g - g[v_r]$ as $g_+$ and $g_-$ respectively:

$$g_+[v] = \begin{cases} g[v] - g[v_r], & \text{if } g[v] \geq g[v_r]. \\ 0, & \text{otherwise.} \end{cases} \tag{25}$$

$$g_-[v] = \begin{cases} |g[v] - g[v_r]|, & \text{if } g[v] \leq g[v_r]. \\ 0, & \text{otherwise.} \end{cases} \tag{26}$$

Now

$$\lambda_2 = \frac{\displaystyle\sum_{u,v \in V} (g[u] - g[v])^2 w_{u,v}}{\displaystyle\sum_v g[v]^2 d_v \tau_v} \geq \frac{\displaystyle\sum_{u,v \in V} (g_+[u] - g_+[v])^2 w_{u,v} + (g_-[u] - g_-[v])^2 w_{u,v}}{\displaystyle\sum_v (g_+[v]^2 + g_-[v]^2) d_v \tau_v}$$

$$\geq \min \left[ \frac{\displaystyle\sum (g_+[u] - g_+[v])^2 w_{u,v}}{\displaystyle\sum_v g_+[v]^2 d_v \tau_v}, \frac{\displaystyle\sum (g_-[u] - g_-[v])^2 w_{u,v}}{\displaystyle\sum_v g_-[v]^2 d_v \tau_v} \right].$$

Without loss of generality, we assume the first ratio is at most the second ratio, and will mostly focus on the vertices $\{v_1, \ldots, v_r\}$ in the first "half" of the graph in the analysis below. Thus,

$$\lambda_2 \geq \frac{\displaystyle\sum_{u,v}(g_+[u]-g_+[v])^2 w_{u,v}}{\displaystyle\sum_v g_+[v]^2 d_v \tau_v} \geq \frac{\left(\displaystyle\sum_{u,v}(g_+^2[u]-g_+^2[v])w_{u,v}\right)^2}{\left(\displaystyle\sum_v g_+[v]^2 d_v \tau_v\right)\left(\displaystyle\sum_{u,v}(g_+[u]+g_+[v])^2 w_{u,v}\right)}$$

which follows from the Cauchy-Schwartz inequality.

We now separately analyze the numerator and denominator. To bound the denominator, we will use the following property of $\tau_i$: because $\mathcal{L}$ is properly scaled, $\tau_i \geq 1$ for all $i \in V$. Therefore,

$$\sum_{u,v}(g_+[u]+g_+[v])^2 w_{u,v} \leq \sum_{u,v} 2(g_+^2[u]+g_+^2[v])w_{u,v}$$

$$= 2\sum_{u \in V} g_+^2[u]d_u \leq 2\sum_{u \in V} g_+^2[u]d_u \tau_u.$$

Hence, the denominator is at most

$$2\left(\sum_{u \in V} g_+^2[u]d_u \tau_u\right)^2.$$

To bound the numerator, we consider subsets of vertices $S_i = \{v_1, \ldots, v_i\}$ for all $1 \leq i \leq r$ and define $S_0 = \emptyset$. First note that

$$\text{vol}_{\mathcal{L}}(S_i) - \text{vol}_{\mathcal{L}}(S_{i-1}) = d_{v_i} \tau_{v_i}. \tag{27}$$

By the definition of $\phi_{\mathcal{L}}(G)$, we know $\phi_{\mathcal{L}}(G) \leq \min_i h_{\mathcal{L}}(S_i)$ for all $1 \leq i \leq r$, where recall the function $h_S(\mathcal{L})$ is defined by Eq. (21). Since $\text{vol}_{\mathcal{L}}(S_i) \leq \text{vol}_{\mathcal{L}}(\bar{S}_i)$ for all $1 \leq i \leq r$, we have

$$\text{cut}(S_i, \bar{S}_i) \geq \phi_{\mathcal{L}} \cdot \text{vol}_{\mathcal{L}}(S_i). \tag{28}$$

By orienting vertices according to $v_1, \ldots, v_n$, we can express the numerator

$$\text{Num} = \left(\sum_{u,v}(g_+^2[u]-g_+^2[v])w_{u,v}\right)^2$$

$$= \left(\sum_{i<j}\left(\sum_{k=0}^{j-i-1} g_+^2[v_{i+k}] - g_+^2[v_{i+k+1}]\right)w_{v_i,v_j}\right)^2.$$

Rewrite the difference as a telescoping series

$$= \left(\sum_{i=1}^{n-1}\left(g_+^2[v_i]-g_+^2[v_{i+1}]\right)\cdot\text{cut}(S_i,\bar{S}_i)\right)^2$$

Collecting $(v_i, v_{i+1})$ terms

$$\geq \left( \sum_{i=1}^{n-1} \left( g_+^2[v_i] - g_+^2[v_{i+1}] \right) \cdot \phi_{\mathcal{L}} \cdot \text{vol}_{\mathcal{L}}(S_i) \right)^2$$

By Eq. (28)

$$= \phi_{\mathcal{L}}^2 \cdot \left( \sum_{i=1}^{n} g_+^2[v_i] \cdot (\text{vol}_{\mathcal{L}}(S_i) - \text{vol}_{\mathcal{L}}(S_{i+1})) \right)^2$$

By Eq. (27) and $g_+(v_n) = 0$

$$= \phi_{\mathcal{L}}(G)^2 \cdot \left( \sum_{i=1}^{n} g_+^2[v_i] \cdot d_{v_i} \tau_i \right)^2 .$$

Combining the bounds for the numerator and the denominator, we obtain $\lambda_2 \leq \phi_{\mathcal{L}}^2 / 2$ as stated in the theorem. The upper bound of $\lambda_2$ follows from the same argument for the standard Cheeger inequality. $\quad\square$

## Spectral partitioning for parameterized conductance

The parameterized Cheeger inequality is essential for providing theoretical guarantees for greedy community detection algorithms. In this section, we extend traditional spectral clustering algorithm to the parameterized Laplacian setting.

Given a weighted graph $G = (V, E, A)$ and a operator $\mathcal{L}$, we can use the standard sweeping method in the proof of Theorem 1 to find a partition $(S, \bar{S})$. This procedure is described in Algorithm 1.

Before stating the quality guarantee of the above algorithm, we quickly discuss its implementation and running time. The most expensive step is the computation of the eigenvector $f$ associated with the second smallest eigenvalue of $\mathcal{L}$. While one can use standard numerical methods to find an approximation of this eigenvector—the analysis would depend on the separation of the second and the third eigenvalue of $\mathcal{L}$. Since $\mathcal{L}$ is a diagonally scaled normalized Laplacian matrix, one can use the nearly-linear-time Laplacian solvers (e.g., by Spielman–Teng (*Spielman & Teng, 2004*) or Koutis–Miller–Peng (*Koutis, Miller & Peng, 2010*)) to solve linear systems in $\mathcal{L}$.

Following *Spielman & Teng (2004)*, let us consider the following notion of spectral approximation of $\mathcal{L}$: suppose $\lambda_2(\mathcal{L})$ the second smallest eigenvalue of $\mathcal{L}$. For $\varepsilon \geq 0, \bar{f}$ is an *$\varepsilon$-approximate second eigenvector* of $\mathcal{L}$ if $\bar{f} \perp D_A^{1/2} T^{1/2} \cdot \vec{1}$, and

$$\frac{\bar{f}^T \mathcal{L} \bar{f}}{\bar{f}^T \bar{f}} \leq (1 + \varepsilon) \cdot \lambda_2(\mathcal{L}).$$

The following proposition follows directly from the algorithm and Theorem 7.2 of *Spielman & Teng (2004)* (using the solver from *Koutis, Miller & Peng, 2010*).

## Algorithm 1. Spectral Dynamics Clustering $(G, \mathcal{L})$

**Input**: weighted network: $G = (V, E, A)$, and spreading operator $\mathcal{L}$ defined by the interaction matrix $W$ and the vertex delay factor $T$.

**Output** partition: $(S, \bar{S})$

**Algorithm**

- Find the eigenvector $f$ of $\mathcal{L} = T^{-1/2} D_W^{-1/2} (D_W - W) D_W^{-1/2} T^{-1/2}$ associated with the second smallest eigenvalue of $\mathcal{L}$.

- Let vector $g$ be $g[u] = f[u] / \sqrt{d_{Wu} \tau_u}$.
- Order the vertices of $G$ into $(v_1, \ldots, v_n)$ such that $g[v_1] \geq g[v_2] \geq \cdots \geq g[v_n]$.
- Sweeping: for each $S_i = \{v_1, \ldots, v_i\}$, compute

$$h_{\mathcal{L}}(S_i) = \frac{\text{cut}(S_i, \bar{S}_i)}{\min\left(\text{vol}_{\mathcal{L}}(S_i), \text{vol}_{\mathcal{L}}(\bar{S}_i)\right)}.$$

- Output the $S_i$ with the smallest $h_{\mathcal{L}}(S_i)$.

**Proposition 1** *For any interaction graph $G = (V, E, W)$ and vertex scaling factor $T$, and $\varepsilon, p > 0$, with probability at least $1 - p$, one can compute an $\varepsilon$-approximate second eigenvector of operator $\mathcal{L}$ in time*

$$O\big(|E| \log n \log \log n \log(1/p) \log(1/\varepsilon)/\varepsilon\big).$$

To use this spectral approximation algorithm (and in fact any numerical approximation to the second eigenvector of $\mathcal{L}$) in our spectral partitioning algorithm for the dynamics, we will need a strengthened theorem of Theorem 1.

**Theorem 2. (Extended Cheeger Inequality with Respect to Rayleigh Quotient):** *For any interaction graph $G = (V, E, W)$ and vertex scaling factor $T$, (whose diagonals are $(\tau_1, \ldots, \tau_n)$), for any vector $u$ such that $u \perp D_A^{1/2} T^{1/2} \cdot \overrightarrow{1}$, if we order the vertices of $G$ into $(v_1, \ldots, v_n)$ such that $g[v_1] \geq \cdots \geq g[v_n]$, where $g = (DT)^{-1/2} \cdot u$ then*

$$\frac{(\min_i h_{\mathcal{L}}(S_i))^2}{2} \leq \frac{u^T \mathcal{L} u}{u^T u},$$

*where $\mathcal{L} = T^{-1/2} D_W^{-1/2} (D_W - W) D_W^{-1/2} T^{-1/2}$ and $S_i = \{v_1, \ldots, v_i\}$.*

**Proof.** The theorem follows directly from the proof of Theorem 1 if we replace vector $f$ (the eigenvector of associated with the second smallest eigenvalue of $\mathcal{L}$) by $u$. This theorem is the analog of a theorem by *Mihail (1989)* for Laplacian matrices. $\square$

The next theorem then follows directly from Proposition 1, Theorem 2 and the definition of $\varepsilon$-approximate second eigenvector of $\mathcal{L}$ that provide a guarantee of the quality of the algorithm of this subsection.

**Table 3** Networks studied in this paper and their properties, including number of vertices and edges, diameter, clustering coefficient, and the number of communities, if known.

| Network | #vertices | #edges | Diameter | Clustering | #communities |
|---|---|---|---|---|---|
| Zachary's Karate Club | 34 | 78 | 5 | 0.588 | 2 |
| College Football | 115 | 613 | 4 | 0.403 | 12 |
| House of Representatives | 434 | 51,033 | 4 | 0.882 | 2 |
| Political Blogs | 1,490 | 16,714 | 9 | 0.21 | 2 |
| Facebook Egonets | 4,039 | 88,234 | 17 | 0.303 | N/A |
| Power Grid | 4,941 | 6,594 | 46 | 0.107 | N/A |

**Theorem 3.** *For any interaction graph $G = (V, E, \boldsymbol{W})$ and vertex delay factor $\boldsymbol{T}$, (whose diagonals are $(\tau_1, \ldots, \tau_n)$), one can compute in time*

$$O(|E| \log n \log \log n \log(1/\varepsilon)/\varepsilon)$$

*a partition $(S, \bar{S})$ such that*

$$h_{\mathcal{L}}(S) = \frac{\displaystyle\sum_{v \in S, u \in \bar{S}} w_{u,v}}{\min\left(\displaystyle\sum_{v \in S} d_{\boldsymbol{W}v} \tau_v, \sum_{v \in \bar{S}} d_{\boldsymbol{W}v} \tau_v\right)} \leq \sqrt{2(1+\varepsilon)\lambda_2(\mathcal{L})}$$

*where $\boldsymbol{T}^{-1/2} \boldsymbol{D}_{\boldsymbol{W}}^{-1/2} (\boldsymbol{D}_{\boldsymbol{W}} - \boldsymbol{W}) \boldsymbol{D}_{\boldsymbol{W}}^{-1/2} \boldsymbol{T}^{-1/2}$, $w_{u,v}$ is the $(u, v)^{th}$ entry of the interaction matrix $\boldsymbol{W}$, and $\lambda_2(\mathcal{L})$ is the second smallest eigenvalue of $\mathcal{L}$. Consequently,*

$$h_{\mathcal{L}}(S) \leq 2\sqrt{(1+\varepsilon)\phi_{\mathcal{L}}(G)} = 2(1+\varepsilon) \sqrt{\min_{S^* \in V} \frac{\displaystyle\sum_{v \in S^*, u \in \bar{S}^*} w_{u,v}}{\min\left(\displaystyle\sum_{v \in S^*} d_{\boldsymbol{W}v} \tau_v, \sum_{v \in \bar{S}^*} d_{\boldsymbol{W}v} \tau_v\right)}}.$$

## EXPERIMENTS

We demonstrate that different dynamic processes can lead to divergent views of network structure in several well studied real-world networks. These networks come from different domains and embody a variety of dynamical processes and interactions, from real-world friendships (Zachary karate club (*Zachary, 1977*)), to online social networks (Facebook (*McAuley & Leskovec, 2012*)), to electrical power distribution (Power Grid (*Watts & Strogatz, 1998*)), to co-voting records (House of Representatives (*Poole, 2012*)) and hyperlinked weblogs on US politics (Political Blogs (*Adamic & Glance, 2005*)), to games played between NCAA football teams (*Girvan & Newman, 2002*). Table 3 lists these networks and their properties. We treat all as undirected networks.

To compare the different perspectives on network structure obtained under the parameterized Laplacian framework, we study the centrality and sweep profiles calculated using the four dynamic operators defined in 'Special Cases.' The *centrality profile* gives the parameterized centrality of each vertex under a given operator. To improve visualization, vertices are ordered by their centrality according to the normalized Laplacian and then rescaled to fall within the same range. Thus, only relative differences in centrality are relevant. The *sweep profile* is similar to the community profile used in *Leskovec et al. (2008)* to study network partitioning. Community profile shows the conductance of the best bisection of the network into two communities of size $k$ and $N-k$ as $k$ is varied. They found that community profiles of real-world networks reveal a ''core and whiskers'' organization, with a large core and many small peripheral communities, or whiskers, loosely connected to the core. In contrast, sweep profile gives the parameterized conductance Eq. (21) of a bisection of the network into communities of size $k$ and $N-k$ using Algorithm 1, not necessarily the best bisection. To improve visualization, we rescale sweep profiles to lie within the same range.

In addition to the sweep profile, we also visualize the best bisection obtained using Algorithm 1 (which corresponds to the minimum of the sweep profile). The visualizations are created using network layout that combines ''Yifan Hu'' and ''Force Atlas'' algorithms from the Gephi software package (*Bastian, Heymann & Jacomy, 2009*). Nodes in the same partition have the same color. We also compare with the ground truth communities, where possible, and report accuracy of the comparison.

### Zachary's Karate Club

The first network we study is a social network consisting of 34 members of a karate club in a university, where undirected edges represent friendships (*Zachary, 1977*). This well-studied network is made up of two assortative blocks centered around the instructor and the club president, each with a high degree hub and lower-degree peripheral vertices. With a simple community structure, this network often serves as a benchmark for community detection algorithms. Its centrality and sweep profiles identified by each operator are shown in Fig. 3. The visualizations show the best bisection of the network obtained by each operator, and the last visualization, which gives the ground truth communities.

Just as many other community detection algorithms, the four parameterized Laplacians give almost identical optimal bisections of this simple network, all of which are close to the ground truth communities, with accuracies ranging from 94.1% to 97.1% (Fig. 3G). Furthermore, their centrality and sweep profiles are very similar as well (Figs. 3A and 3B). This is a excellent example showing that most good community measures capture the same fundamental idea of communities, those well-interacting subsets of vertices with relatively sparse connection in between. They do differ, however, in finer details of their mathematical definitions, as we will see in more complicated networks in the following subsections.

### College football

The second network represents American football games played between Division IA colleges during the regular season in Fall 2000 (*Girvan & Newman, 2002*), where two vertices (colleges) are linked if they played in a game. Following the structure of the divisions,

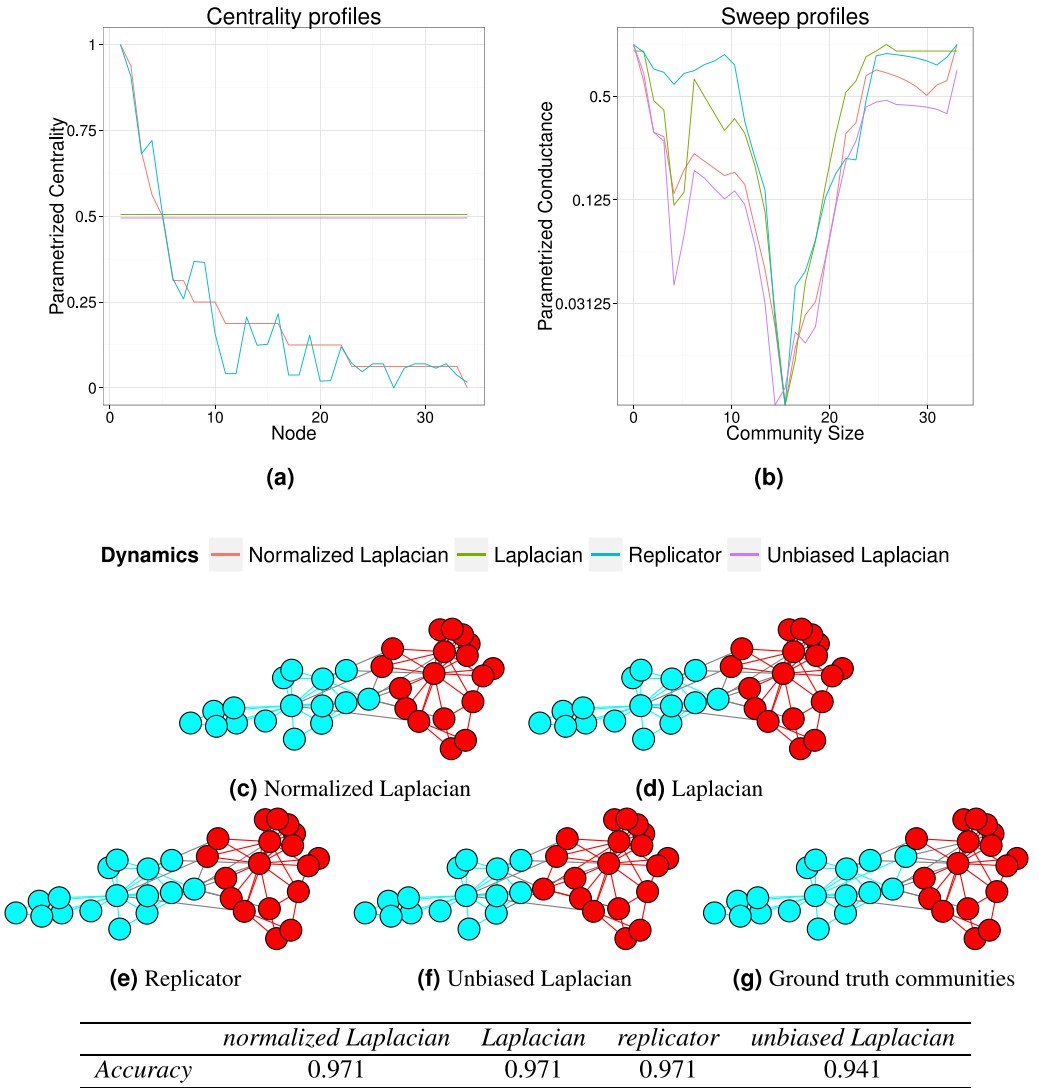

**(a)** Centrality profiles

**(b)** Sweep profiles

**Dynamics** — Normalized Laplacian — Laplacian — Replicator — Unbiased Laplacian

**(c)** Normalized Laplacian

**(d)** Laplacian

**(e)** Replicator

**(f)** Unbiased Laplacian

**(g)** Ground truth communities

| | normalized Laplacian | Laplacian | replicator | unbiased Laplacian |
|---|---|---|---|---|
| *Accuracy* | 0.971 | 0.971 | 0.971 | 0.941 |

**Figure 3 Analysis of the Karate Club network.** Centrality and sweep profiles and optimal bisections of Zachary's Karate Club identified by the four special cases of the parameterized Laplacian. The table reports accuracy of the bisection.

the network naturally breaks up into 12 smaller conferences, roughly corresponding to the geographic locations of colleges. Most games are played within each conference which leads to densely connected local clusters. Its centrality and sweep profiles and visualizations of optimal bisections under each operator are shown in Fig. 4.

The centrality profiles show heavy tailed distributions, which corresponds to evenly spread out degrees across the network Fig. 4A. This is consistent with the reality of the network, where every football team plays roughly the same number of games each season.

Unlike Karate Club, College Football starts to give us different community divisions under different dynamic operators. Most operators lead to a balanced east–west bisection (Figs. 4C, 4D and 4F). This division is mostly consistent (around 95%) with the bisection produced by merging 6 conferences (label 0,1,4,5,6,9 for the east cluster) on each side, as

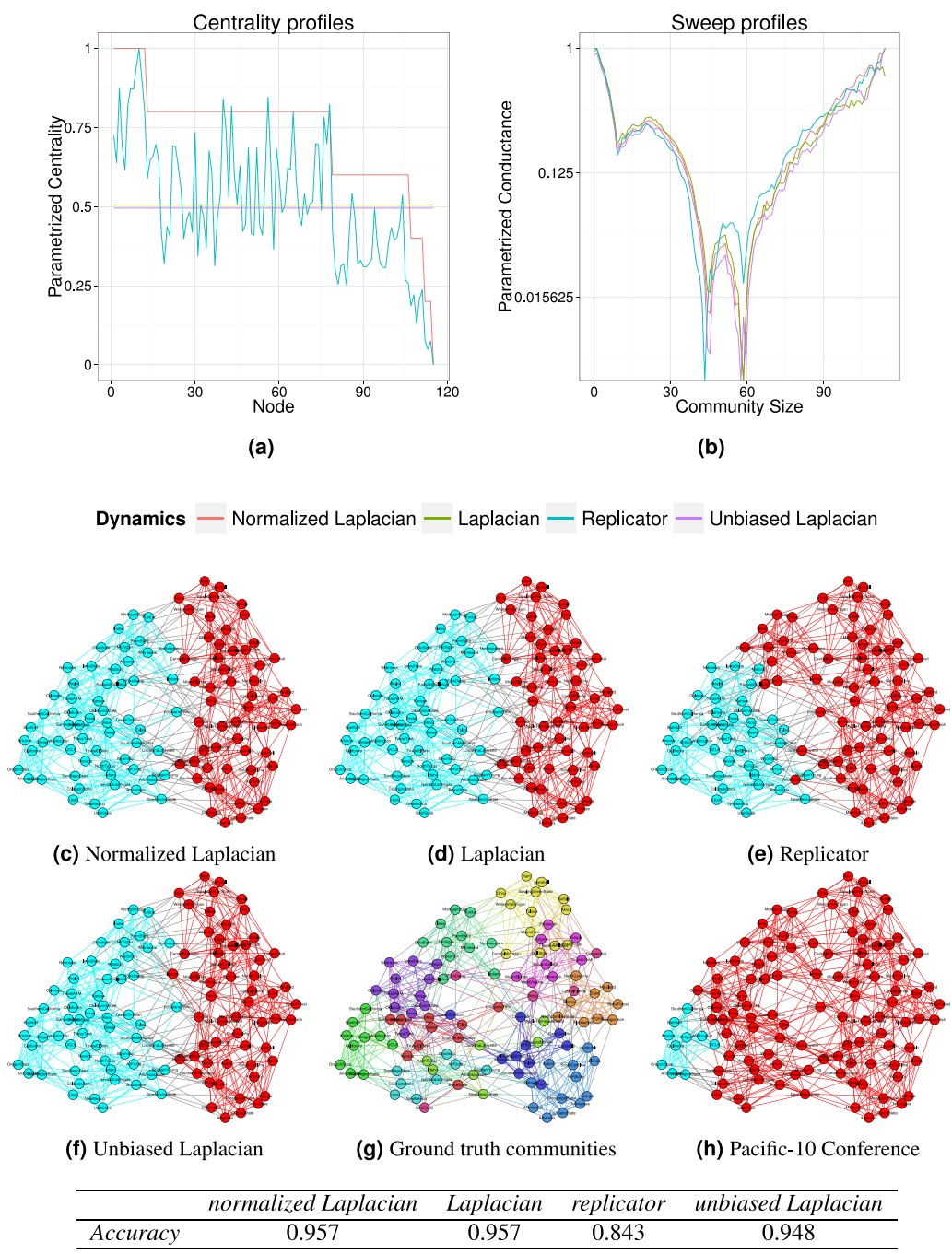

**Figure 4** **Analysis of the College Football network.** Centrality and sweep profiles and optimal bisections of the College Football network identified by the four special cases of the parameterized Laplacian.

illustrated by the accuracy numbers. The replicator, however, places the "swing" Big Ten Conference (contains mostly colleges in the midwest) into the east cluster (Fig. 4E). Upon further investigation, we discovered that while both bisections have almost the same cross community edges, the seemingly more balanced division does lead to a slight imbalance in

terms of links within each community. The the parametrized centrality under the replicator magnified this imbalance, ultimately pushed the "swing" conference to the east side.

In fact, the sweep profile Fig. 4B clearly shows that all four special cases actually see both bisections as plausible solutions, with closely matched local optima. This phenomenon where different dynamics agrees on multiple local optima but favor different ones as the global solution is a repeating theme in the following examples. This means that while different special cases of the parameterized conductance can differ in finer details, they will agree on strong community structures that impact all dynamics in similar ways. Figure 4H further illustrates the point. All four special cases here agree on the first local optimum in the sweep profiles, and this local cluster corresponding to the Pacific 10 conference (it later becomes the Pacific 12).

## House of representatives

The House of Representatives network is built from the voting records of the members of 98th United States House of Representatives (*Poole, 2012*). Unlike previously studied variants (*Waugh et al., 2009*), here we use a special version taking into account all 908 votes. The resulting network has a dense two-party structure with 166 Republicans and 268 Democrats. This network better differentiates some of the dynamics under our framework. Its centrality and sweep profiles and visualizations of optimal bisections under each operator are given below.

The "House of Representatives" network is an excellent example of how centralities and communities are closely related under our framework. First, the centrality profile of this network looks similar to that of the College Football, but quite different from the other networks in Table 3. Because we have taken into account all votes, this network is very densely connected, and its degree distribution also has a heavy tail as demonstrated by the red curve in Fig. 5A.

Since the degree distribution is relatively uniform, we expect the change of the cut size (numerator) in Eq. (21) to be relatively small. The exception here is the optimal bisection produced by the regular Laplacian (Fig. 5D), which is most prone to "whiskers," leading to a low accuracy of 38.5%. For the other three special cases, the volume balance (denominator) is the determining factor in communities measures, and all produce fairly "balanced" bisections according their own parametrized volume measures.

Another observation is that centrality measures disagree about importance of vertices. In particular, centralities given by the normalized Laplacian might differ from those of the unbiased Laplacian by the degree, but given its relative uniform distribution, leads to almost identical optimal bisections (Figs. 5C and 5F). The replicator, on the other hand, scales vertex centrality according to eigenvector centralities, which places more volume to the high degree vertices on the cyan cluster. The resulting optimal bisection is thus shifted to the right to balance volumes (Fig. 5E). In this case, the ground truth aligns closer to the formers with over 90% accuracies as Democrats dominated the 98th Congress.

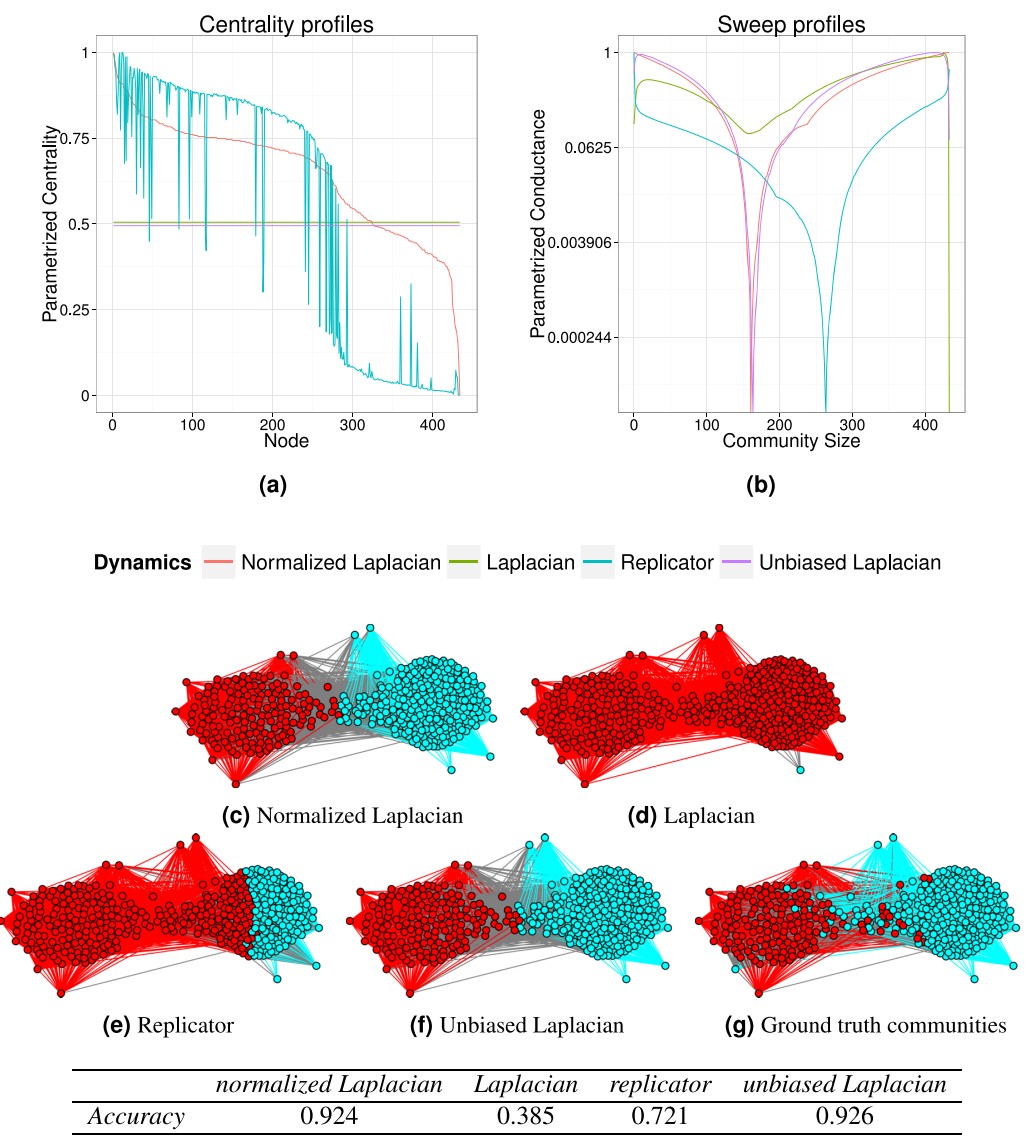

**Figure 5** **Analysis of the House of Representatives covoting network.** Centrality and sweep profiles and optimal bisections of the House of Representatives network identified by the four special cases of the parameterized Laplacian.

## Political Blogs

The next example is the political blogs network (*Adamic & Glance, 2005*). Here we focus on the largest component, which consists of 1,222 blogs and 19,087 links between them. The blogs have known political leanings, and were labeled as either liberal or conservative. The network is assortative and has a highly skewed degree distribution. Its centrality and sweep profiles and visualizations of optimal bisections under each special case dynamic are given below.

The Political Blogs network demonstrates a pitfall of many commonly used community quality measures. Many real world networks have a skewed degree distributions, which often corresponds to a "core-whiskers" (also known as core–periphery) structure. As

shown in _Leskovec et al. (2008)_, such structures have "whisker" cuts that are so cheap that balance constrains can be effectively ignored. The same happened here for three of our special cases, whose optimal bisections are highly unbalanced. They have below 50% accuracies when compared to the ground truth.

Unlike the House of Representatives, community measure in Political Blogs is dominated by the cut size (numerator). In particular, both the normalized Laplacian and the Laplacian share the same cut size measures, give the same solution (Figs. 6C and 6D), despite their differences in volume/centrality measures (see curves in Fig. 6A). The unbiased Laplacian produces a different whisker cut, because it has a reweighed cut size measure (Fig. 6F). Further investigation reveals that the unbiased Laplacian cuts off a whisker from two highly connected vertices, which according to Eq. (21) greatly reduces the cut size.

The exception here is the replicator operator (Fig. 6E). By reweighing the adjacency matrix by eigenvector centralities, the parameterized volume measure now considers highly connected vertices near the core to be even more important (see the red curve in the centrality profile). The difference in parameterized volume is now too drastic to be ignored. As a result, replicator does not fall for the "whisker" cuts and produces balanced communities with a respectable accuracy of 95.3%.

## Facebook Egonets

The Facebook Egonets dataset was collected using a Facebook app (_McAuley & Leskovec, 2012_). Each egonet is a local network that consists of one user's Facebook "friends" that represent that user's social circles. We use the combined network that merges all egonets. This network has many typical social network properties, including a heavy tailed degree distribution. However, it also differs from traditional social networks because of the sampling bias in the data collection process, leading to lower clustering coefficient and a bigger diameter than what one might expect. Its centrality and sweep profiles and visualizations of optimal bisections under each special case dynamic are given below.

As with Political Blogs, the overall multi-core structure leads to unbalanced bisections. Due to its bigger size and an even more heterogeneous degree distribution (Fig. 7A), all four special cases of the parameterized Laplacian fall for local clusters, each in a different fashion. Again, the ordinary Laplacian finds the smallest local community with the minimal cut size of 17 links (Fig. 7D). In contrast, the unbiased Laplacian which has the same volume measure, finds a superset of vertices as the optimal cut, with 40 inter community edges (Fig. 7F). The normalized Laplacian measures cut sizes the same way as the Laplacian, but its different volume measure leads to a much more balanced cut (Fig. 7C). Last but not least, the replicator finds a local core structure with an average degree of 85.7 (Fig. 7E). This is consistent with what we observed on House of Representatives, where the eigenvector centrality places more volume in the cyan cluster, and the resulting cut is actually much more balanced than it looks.

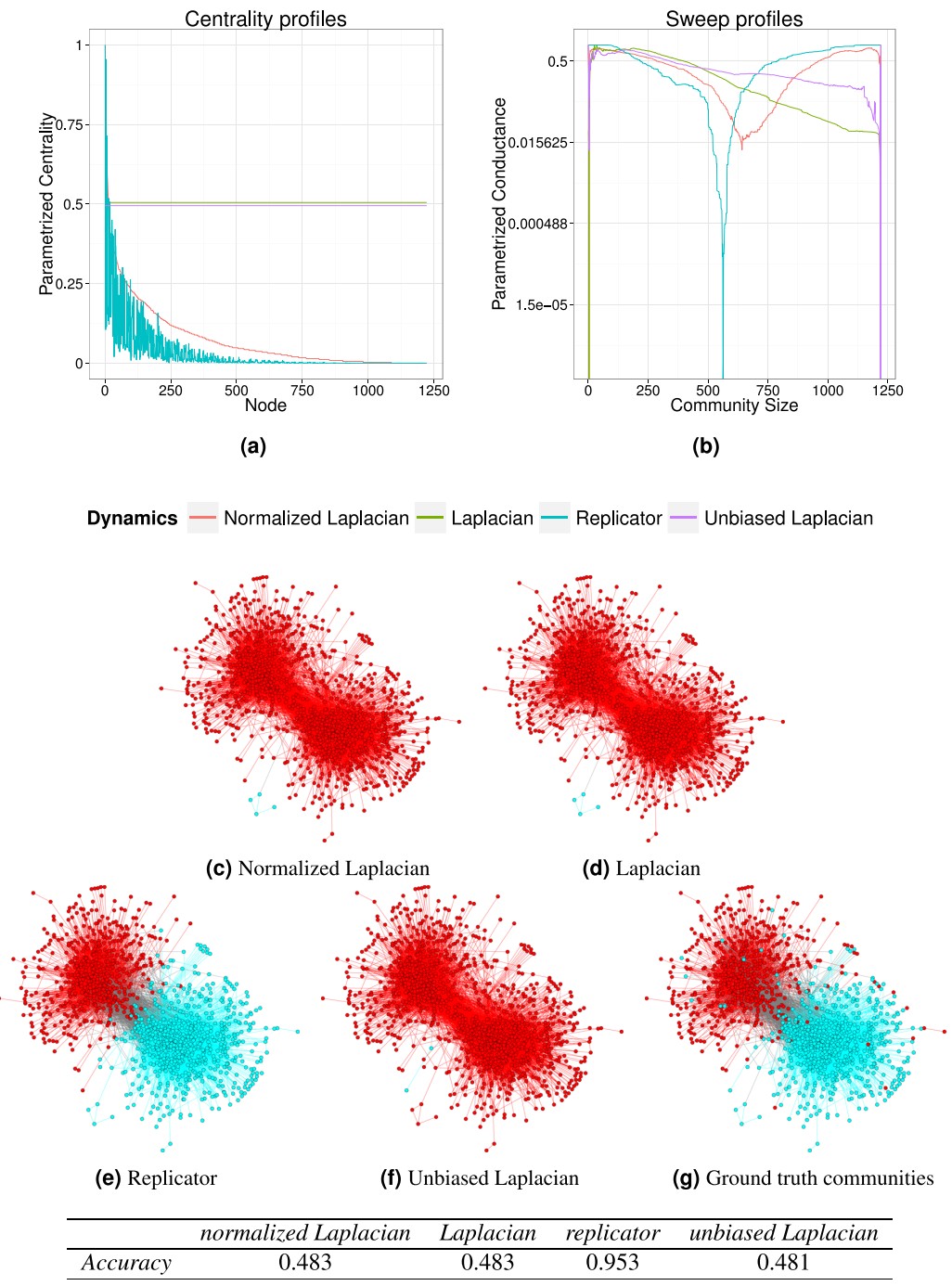

| | normalized Laplacian | Laplacian | replicator | unbiased Laplacian |
|---|---|---|---|---|
| *Accuracy* | 0.483 | 0.483 | 0.953 | 0.481 |

**Figure 6** **Analysis of the Political Blogs network.** Centrality and sweep profiles and optimal bisections of the Political Blogs network identified by the four special cases of the parameterized Laplacian.

## Power grid

The last example is an undirected, unweighted network representing the topology of the western United States power grid (*Watts & Strogatz, 1998*). Among the six datasets in Table 3, Power Grid is the largest network in terms of the number of vertices. However, it

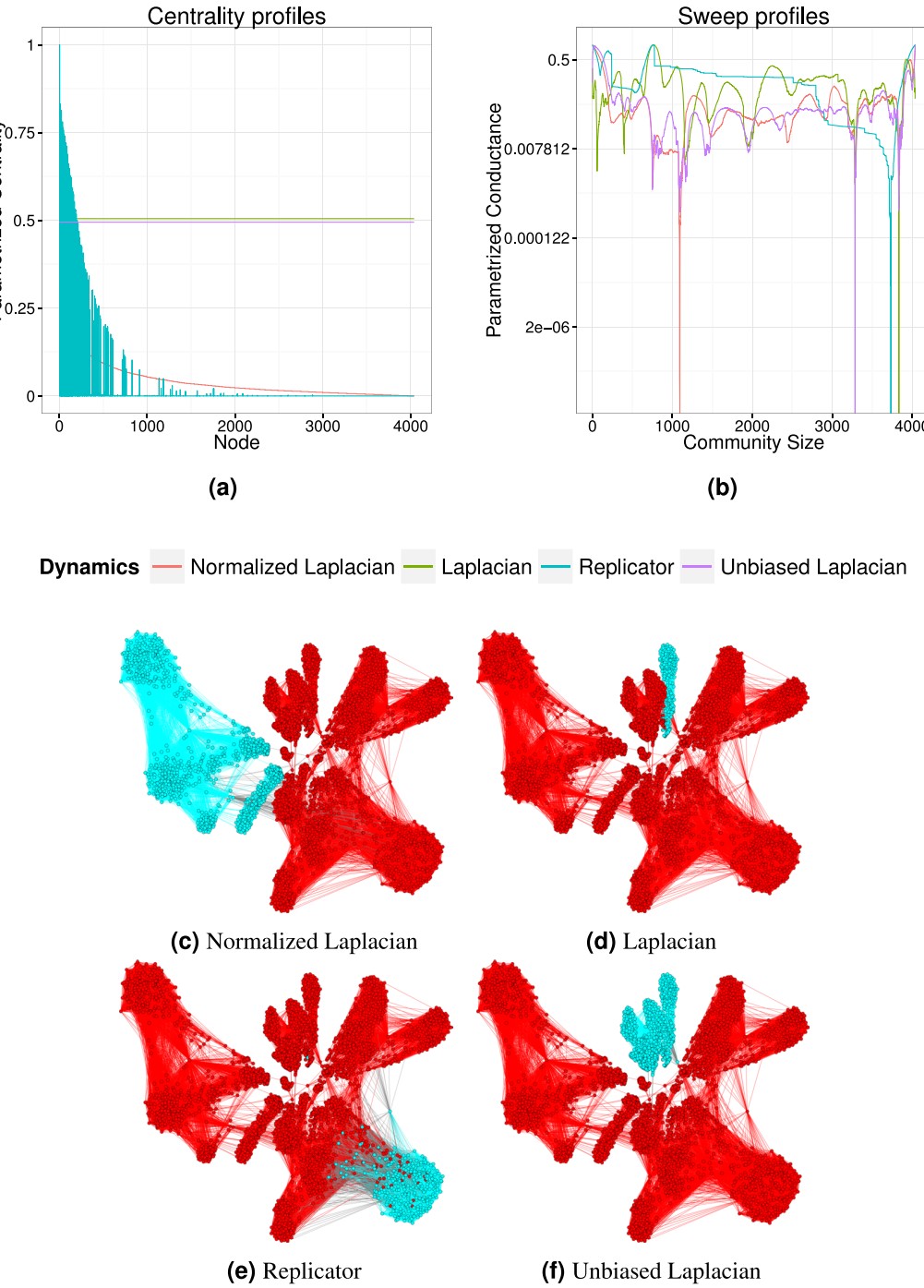

**Figure 7** **Analysis of the Facebook Egonets network.** Centrality and sweep profiles and optimal bisections of the Facebook Egonets network identified by the four special cases of the parameterized Laplacian.

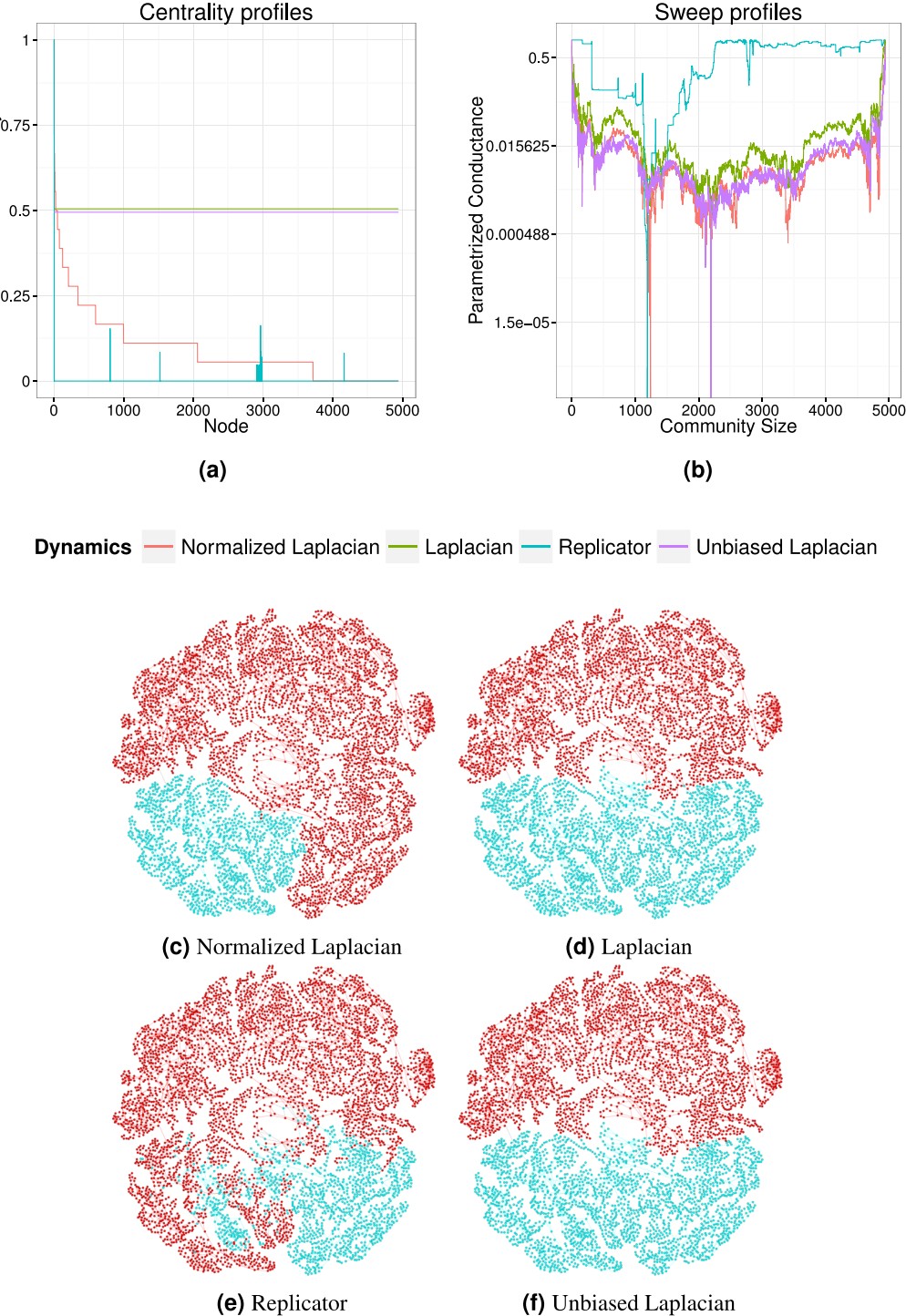

**Figure 8  Analysis of the Power Grid network.** Centrality and sweep profiles and optimal bisections of the Power Grid network identified by the four special cases of the parameterized Laplacian.

is extremely sparse with an average degree of 2.67, leading to a homogeneous connecting pattern across the whole network without core–periphery structure. Its centrality and sweep profiles and visualizations of optimal bisections are given below.

The long tails of the centrality profiles indicate existence of high degree vertices, or hubs Fig. 8A. However, as the visualizations of network bisection show, these hubs do not usually link to each other directly, resulting in negative degree assortativity (*Newman, 2003*). This is consistent with the geographic constrains when designing a power grid, as the final goal is to distribute power from central stations to end users. These important difference in overall structure prevented core or whiskers from appearing, and changes how different dynamics behave on Power Grid.

Replicator, which demonstrated the most consistent performance on social networks with core–periphery structure, performs the worst on bisecting the Power Grid. In fact, the visualization shown in Fig. 8E is obtained by manually fixing negative eigenvector centrality entries in 'Replicator' (the numeric error comes from the extreme sparse and ill-conditioned adjacency matrix).

The other three special cases all give reasonable results. Laplacian and unbiased Laplacian share the same volume measure, and they have nearly identical solutions with well balanced communities (Figs. 8D and 8F). Their different cut size measures only lead to slightly different boundaries thanks to the homogeneous connecting pattern. Normalized Laplacian share the same cut size measure with the regular Laplacian, and its volume balance is usually more robust on social networks with core-whisker structures. On Power Grid, however, it opts for a smaller cut size at the cost of volume imbalance (Fig. 8C). It turns out the volume of the cyan cluster is compensated by its relative high average degree.

## CONCLUSION

The parameterized Laplacian framework presented in this paper can describe a variety of dynamical processes taking place on a network, including random walks and simple epidemics, but also new ones, such as one captured by the unbiased Laplacian. We extended the relationships between the properties of centrality, community-quality measures and properties of the Laplacian operator, to this more general setting. Each dynamical process has a stationary distribution that gives centrality of vertices with respect to that process. In addition, we show that the parameterized conductance with respect to the dynamical process is related to the eigenvalues of the operator describing that process through a Cheeger-like inequality. We used these relationships to develop efficient algorithm for spectral bisection.

The parameterized Laplacian framework also provides a tool for comparing different dynamical processes. By making the dynamics explicit, we gain new insights into network structure, including who the central nodes are and what communities exist in the network. By connecting the operators using standard linear transformations, we discovered an equivalence among different dynamical systems. In the future, we plan to investigate their differences based on how the vertex state variables change during the evolution of the dynamic process. In the analysis of massive networks, it is also desirable to identify

subsets of vertices whose induced sub-graphs have "enough" community structure without examining the entire network. *Chung (2007)* and *Chung (2009)* derived a local version of the Cheeger-like inequality to identify random walk-based local clusters. Similarly, our framework can be adapted to such local clustering procedures.

While our framework is flexible enough to represent several important types of dynamical processes, it does not represent all possible processes, for example, those processes that even after a change of basis, do not conserve the total volume. In order to describe such dynamics, an even more general framework is needed. We speculate, however, that the more general operators will still obey the Cheeger-like inequality, and that other theorems presented in this paper can be extended to these processes.

### Funding

This work was partly supported by grants NSF CIF-1217605, AFOSR-MURI FA9550-10-1-0569, AFRL FA-8750-12-2-0186, DARPA W911NF-12-1-0034, NSF CCF-0964481, and NSF CCF-1111270. The funders had no role in study design, data collection and analysis, decision to publish, or preparation of the manuscript.

### Grant Disclosures

The following grant information was disclosed by the authors:
NSF: CIF-1217605, CCF-0964481, CCF-1111270.
AFOSR-MURI: FA9550-10-1-0569.
AFRL: FA-8750-12-2-0186.
DARPA: W911NF-12-1-0034.

### Competing Interests

Kristina Lerman is an Academic Editor for PeerJ Computer Science. Rumi Ghosh is an employee of Robert Bosch LLC.

### Author Contributions

- Xiaoran Yan performed the experiments, analyzed the data, wrote the paper, prepared figures and/or tables, performed the computation work.
- Shang-hua Teng and Kristina Lerman conceived and designed the experiments, wrote the paper, reviewed drafts of the paper.
- Rumi Ghosh conceived and designed the experiments, performed the computation work.

### Data Availability

The research in this article did not generate any raw data.

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
