# Peer review of "Capturing the interplay of dynamics and networks through parameterizations of Laplacian operators"

_PeerJ Computer Science, doi:10.7717/peerj-cs.57_

## Round 0.1 · original submission · Major Revisions

Please see the attached reviews. Both reviewers are experts on stochastic processes in networks, and both have expressed very similar concerns. The main concern is that many pertinent references have not been cited. These references should be incorporated and the current work should be contrasted with these to bring out what exactly is novel in the paper. Both have also expressed concern that standard concepts in Markov chains are explained in great detail. This aspect needs to be fixed as well. Please also address the notational and terminology issues raised by the reviewers.

Reviewer 1 ·

Basic reporting

The theoretical novelty in this paper is to extend the ideas of network centrality, set conductance and its relation to the spectral gap, as defined under the basic random walk on the network, to a more general family of stochastic dynamic processes on the network. These processes are characterized via a generalization of the Laplacian, parametrized by three quantities - scaling and reweighting matrices $T$ and $W$ (which correspond to biasing the walk by increasing the self-loop probability/delay of individual states, or adding edge-weights), and an exponent $\rho$ which allows one to view different dynamics (in particular, 'epidemic' models), as similarity transformations of the random walk transition matrix. The paper discusses how existing models fit within this framework, derive an analog of Cheeger's inequality for these settings, and then applies the clustering algorithms to various networks.

The paper was well written, and enjoyable to read. However I have a few issues with this work, in particular with respect to its relation to some existing works, which I would like the authors to address.
Major issues:
* The main ideas in this paper seem to be to extend the ideas of conductance and Cheerger's inequality to stochastic dynamic processes on graphs. However, there is a very well established line of work on Markov chain mixing times, starting from the seminal papers by Jerrum and Sinclair (M. Jerrum and A. Sinclair. Conductance and the rapid mixing property for Markov chains: the approximation of the permanent resolved, 1988) and Lawler and Sokal (G. Lawler and A. Sokal. Bounds on the $\ell_2$ spectrum for Markov chains and Markov processes: a generalization of Cheeger’s inequality, 1988), which establish such results for any general Markov Chain (also see the book by Aldous and Fill on reversible Markov Chains - all the Markov chains considered in this work appear to be reversible, in particular as the underlying graphs are undirected). I am unsure how this paper's contributions go beyond these known results, and would like the authors to add some references to this area, and clarify the differences.
* Also, the authors reference an earlier work of theirs (Ghosh, R., Teng, S.-h., Lerman, K., and Yan, X. (2014). The Interplay Between Dynamics and Networks: Centrality, Communities, and Cheeger Inequality. ) which introduces a parametrized Laplacian framework - although they claim the new framework is different, I could not figure out the differences, albeit after a quick look at the older work. Any difference should be clarified in more detail.

Minor comments:
* In Markov chain literature, P_{ij} is conventionally used to denote the transition probability from $i$ to $j$, and correspondingly, all state probability vectors (usually denote $\pi_t$) are taken to be row vectors (so $\pi_t = \pi_{t-1}P$). It was very confusing for me to read everything transposed, especially as I had not paid close attention to the definitions at the start - it would be very useful if the authors add a footnote early on warning readers of this difference in notational conventions.
* The set of models referred to as epidemic models appear to be very similar to the de Groot consensus process (see the Wikipedia entry, and references therein) - more generally, such processes are typically referred to as consensus, and the term epidemic is used for percolation-style dynamics. It would be useful if the authors clarify this.
* The authors use $\theta$ to denote very different objects - in case of random walks, it is used to denote the probability vector, while in case of consensus processes, it is used to denote 'belief' vectors. I understand that the authors do so to show the similarities between the two - however, I feel this is more confusing, in particular for readers not familiar with these processes.
* There are some small errors and inconsistencies in the writing. A few examples:
- (Pg 4) "the entries of diagonal matrix V is the components of the eigenvector V"
- (Pg 5) "linearized approach to synchronization of different variants of the Kuramoto model" (the Kuramoto model is not defined here)
- (Pg 5) "Let vol(S) = ∑i∈ di"
- (Pg 6) "The smallest achievable such ratio is also known as..." - smallest over subsets
- (Pg 6) "one can then perform a partitioning-based algorithm" - is this a standard term?
- (Pg 6) "the presence of a good cluster implies " - 'good' is undefined at the moment
- (Pg 8) "In fact, we can manipulate the adjacency matrix in any way (we can even come up completely new matrices) as long as the result is still an positive semi-definite and symmetric matrix, for any perceived “interaction graph” of dynamics."
- (Pg 11) "However, the uniform value associated with each vertex is" - this is very non-standard, and seems somewhat forced as a definition.
- (Pg 12) The summation limits are not clearly specified (for i)
- (Pg 12) "any symmetric real matrix has an orthonormal basis V which consists of its eigenvectors" The matrix need not be full rank - this can be more correctly stated.
* The PeerJ instructions indicate that acknowledgements should not include any funding aknowledgement.

Experimental design

Not applicable, as the work does not have any experiments.

Validity of the findings

Not applicable, as the work does not present any new data.

·

Basic reporting

I will write the comments on the criteria as stated on the PeerJ website, in the same order.

There are many grammatical errors and typos. Some typos result in mathematically incorrect statements. For example:
- on page 8 \Theta(0) is left-multiply with a matrix, while throughout the paper it is a column-vector and is right-multiplied with a matrix.
- p. 12 `with a random walk as a standard basis' is mathematically incorrect
- p. 19: `eigenvalue vector'
- p.4: in the second displayed formula \theta_j should be \theta_j(t). A similar typo appears elsewhere as well.
- p.18: `The right hand side of the theorem'

I personally did not like how the citations looked because the authors use a author-year style, but also use the names in the text, so the names are printed double: for example on p.20: `used by Lescovec et al. (Lescovec et al., 2008)'

Introduction implies much greater generaliy of the results than are actually in the paper.

Avrachenkov et al. in a series of papers considered similar parametrization and also discussed \rho=-1/2, 0, 1/2. They also used this approach for community detection, but in a setting of semi-supervised learning, when some representatives of clusters are known. The conductance approach in this paper could be complementary to the approach by Avrachemkov et al. It would be also interesting to compare the results. Definitely this highly relevant work must be properly referenced. E.g.
@inproceedings{avrachenkov2012generalized,
title={Generalized optimization framework for graph-based semi-supervised learning},
author={Avrachenkov, Konstantin and Gon{\c{c}}alves, Paulo and Mishenin, Alexey and Sokol, Marina},
booktitle={Proceedings of SIAM Conference on Data Mining (SDM 2012)},
volume={9},
year={2012},
organization={SIAM}
}
@incollection{avrachenkov2013choice,
title={On the choice of kernel and labelled data in semi-supervised learning methods},
author={Avrachenkov, Konstantin and Gon{\c{c}}alves, Paulo and Sokol, Marina},
booktitle={Algorithms and Models for the Web Graph},
pages={56--67},
year={2013},
publisher={Springer}
}

I find strange the reference to Lambiotte et al. 2011 on page 4. The described process is just a continuous time Markov chain. You can refer to a standard textbook, such as one by Ross or Karlin&Taylor.

I think that on page 10 the authors should not include a citation to their article next to Katz and Brin and Page, because the reference to their own work is rather specific, while the others are more classical and general.

The paper should be drastically shortened and the structure must be improved. In the current version new results start on page 16. Before that, general concepts are explained. Here are my concerns:
- The presented concepts are not new. The \rho-parametrization has been already considered e.g. by Avrachenkov et al. The T-parametrization is a standard approach in continous-time Markov chains. The W-parametrization is just a special case of a weighted graph. An interesting parametrization is a replicator, but it has been already suggested in the authors' previous work.
- The explanation is sometimes confusing. For example, in (21), eigenvectors of which caligraphic L do you mean? Also the formula suggests that z_1 depends on \theta(0), which is obviously not the case. A more clear exposure can be found in the on-line book on Markov Chains by Aldous and Fill, see chapter 3, section 4.
- What is the goal of explaining known results in such length?

Figures are well presented and are interesting.

The submission is mostly self-contained. In a couple of places more explanation is needed:
- Explain what the term `sweep procedure' means.
- Explain what the term `whisker' means.

There are typo/errors in the proof of Theorem 1, but I think those can be fixed.
- Why to use subindex caligraphic L in the definition of the volume? It seems to me that the volume as you define it is the same for any operator.
- Does the proof go thourgh only for the L as defined in the formulation of Theorem 1 or for other special cases as well? Same question about Algorithm 1.
- In the formula under line 396 on page 17 you sum over all pairs (u,v), so each pair appears twice, as (u,v) and as (v,u). Then, with W symmetric, the numerator in the second line is equal to zero.

In Algorithm 1 it is not clear whether only one case of L is used or it is applied to all different cases. Therefore, it is hard to understand where the computations for the four special cases differ in the community detection part. Only conductances are different? Or also the sets S_i? Or also the volume? This should be made clear in Algorithm 1 and in the experimental section.

I had difficulty following the text in Section 6. It is too long, and did not make it clear to me why we observe the differences between different procedures.

I found the figures interesting, and the experiments comprehensive and reproducible.

Notations are often confusing. For example:
- In the proof of Theorem 1, d_i is used instead of d_{W_i} (the authors say `with slight obuse of notations'). This is very confusing because d_i has a different meaning. Besides, in the formulation of Theorem 3, the notation d_{W_i} is used again.
- Caligraphic L is used interchangeably to different operators, it is often not clear which one the authors mean.
- Notation \nu is used for eigenvectors, but also for vertices e.g. in the proof of Theorem 1.
- There are many other instances.

Experimental design

From the math point of view, the original part is the proof Theorem 1.

The level of the results is weaker than the problem statement, which is very general. The experimental results are interesting.

Validity of the findings

If the errors/typos in the proof are fixed and the notations are clarified, I expect the results to be valid.

Additional comments

No comments

---

## Round 0.2 · accepted · Accept

The revisions are satisfactory, and the contributions are explained clearly. Though the contributions build on a previous conference paper, the unification of many previous ideas into one paper makes it worthy of publication.